# Inhibitory Effect of Polyphenols from the Whole Green Jackfruit Flour against α-Glucosidase, α-Amylase, Aldose Reductase and Glycation at Multiple Stages and Their Interaction: Inhibition Kinetics and Molecular Simulations

**DOI:** 10.3390/molecules27061888

**Published:** 2022-03-14

**Authors:** Tejaswini Maradesha, Shashank M. Patil, Khalid Awadh Al-Mutairi, Ramith Ramu, SubbaRao V. Madhunapantula, Taha Alqadi

**Affiliations:** 1Department of Biotechnology and Bioinformatics, School of Life Sciences, JSS Academy of Higher Education and Research, Mysuru 570015, Karnataka, India; tejaswini@jssuni.edu.in (T.M.); shashankmpatil@jssuni.edu.in (S.M.P.); 2Biology Department, Faculty of Science, University of Tabuk, Tabuk 71491, Saudi Arabia; kmutairi@ut.edu.sa; 3Center of Excellence in Molecular Biology and Regenerative Medicine (CEMR, A DST-FIST Supported Center), Department of Biochemistry (A DST-FIST Supported Department), JSS Medical College, JSS Academy of Higher Education and Research, Mysore 570015, Karnataka, India; mvsstsubbarao@jssuni.edu.in; 4Department of Biology, Adham University College, Umm Al-Qura University, Makkah 21955, Saudi Arabia; taqadi@uqu.edu.sa

**Keywords:** whole jackfruit, type 2 diabetes, phenolic acids, caffeic acid, syringic acid, inhibition kinetics, molecular dynamics simulation

## Abstract

For the first time, α-glucosidase, α-amylase, aldose reductase, and glycation at multiple stages inhibitory assays were used to explore the antidiabetic potential of whole unripe jackfruit (peel with pulp, flake, and seed). Two polyphenols (phenolic acids) with strong antihyperglycaemic activity were isolated from the methanol extract of whole jackfruit flour (MJ) using activity-guided repeated fractionation on a silica gel column chromatography. The bioactive compounds isolated were identified as 3-(3,4-Dihydroxyphenyl)-2-propenoic acid (caffeic acid: CA) and 4-Hydroxy-3,5-dimethoxybenzoic acid (syringic acid: SA) after various physicochemical and spectroscopic investigations. CA (IC_50_: 8.0 and 26.90 µg/mL) and SA (IC_50_: 7.5 and 25.25 µg/mL) were identified to inhibit α-glucosidase and α-amylase in a competitive manner with low Ki values. In vitro glycation experiments further revealed that MJ and its components inhibited each stage of protein glycation as well as the generation of intermediate chemicals. Furthermore, CA (IC_50_: 3.10) and SA (IC_50_: 3.0 µg/mL) inhibited aldose reductase effectively in a non-competitive manner, respectively. The binding affinity of these substances towards the enzymes examined has been proposed by molecular docking and molecular dynamics simulation studies, which may explain their inhibitory activities. The found potential of MJ in antihyperglycaemic activity via inhibition of α-glucosidase and in antidiabetic action via inhibition of the polyol pathway and protein glycation is more likely to be related to the presence of the phenolic compounds, according to our findings.

## 1. Introduction

Type 2 diabetes mellitus (T2DM) is a metabolic condition marked by persistently high blood glucose levels (hyperglycaemia). The International Diabetes Federation anticipated that this condition would impact over 783 million individuals by 2045 [1]. Hyperglycaemia causes the production of advanced glycation end-products (AGEs), the accumulation of sorbitol due to overactivation of the polyol pathway, and increased production of free radicals beyond normal physiological control, resulting in retinopathy, nephropathy, and cardiovascular disease [2]. It has been proposed that lifestyle changes, including increased physical activity and eating a diet rich in plant-derived foods (e.g., whole grains, fruits, and vegetables), might prevent 90 percent of T2DM cases [3]. The favourable health benefits of plant-derived products have been attributed to plant phytochemicals such as polyphenolic compounds, as well as vitamins, minerals and dietary fibre. Polyphenols, in this regard, have been shown to reduce the severity of T2DM symptoms (such as fasting and postprandial hyperglycaemia) by blocking disaccharidases (such as α-amylase and α-glucosidase) in the intestine lumen [4].

*Artocarpus heterophyllus* L. (Jackfruit) belongs to the mulberry family (Moraceae). Although it is native to India, it can also be found in Malaysia, Sri Lanka, India, Indonesia, the Philippines, Burma, Pakistan, Burma, and China. It is a high-yielding crop that bears fruit all year, with the best months being June and December [5]. The leaves and roots have traditionally been used to cure wounds, dermatosis, anaemia, diarrhoea, and asthma. The jackfruit pulp, leaf, root, and bark have been evaluated for their phytochemical and pharmacological properties [6]. Jackfruit pulp extract has proven anti-inflammatory properties by inhibiting the formation of nitric oxide (NO) and prostaglandin E2 (PGE2). Similarly, jackfruit leaf extracts have antioxidant properties and can reduce hyperglycaemia and hyperlipidaemia. Phenolic acids, organic acids, flavonoids, triterpenes, carotenoids, stilbenes, and sterols have all been found in various parts of this plant, including root, bark, leaf, and fresh fruit, with prenyl flavonoids being the most abundant [7].

Fruit axis (fruit core), the pulp (middle fleshy edible bulb), flakes (the exterior wrapped silk-like component of pulps), and peel make up the entire jackfruit (the exterior horny and non-edible part). With the exception of the fruit axis and peel, jackfruit is eaten as a fresh fruit when ripe and as a vegetable when unripe [8]. The pulp is also used to make fruit snacks, fruit juice, and fruit wine, among the others. The peel, which accounts for around 46% of the fruit, is underused and is typically dumped as trash or fertiliser [5]. Furthermore, there is no study on the phytochemical components and bioactivity of the whole unripe jackfruit. Understanding the phytochemical components and biological activity of the green and unripe jackfruit is critical for determining its application, potential and value. In this regard, the primary objective of this study is to evaluate the efficacy of whole unripe jackfruit extract in inhibiting α-glucosidase and α-amylase, which are the key enzymes causing hyperglycemia; as well as aldose reductase and protein glycation, which cause numerous diabetes complications. Glycation indicators at various stages of glycation, such as fructosamines (early glycation), protein carbonyls (middle-stage), and AGEs (late-stage), were also studied. As a result, the current work was a bioassay-guided separation of the active antihyperglycaemic compounds from the methanol extract of whole jackfruit flour, followed by in silico approaches to estimate druggability and probable target proteins to which these compounds bind (Appendix A). Future experimental work will hopefully be prioritised based on the findings of this study, resulting in the successful creation of novel antihyperglycaemic medicines.

## 2. Materials and Methods

The whole mature and unripe jackfruits (94–101 days) were obtained during May 2019 from the Chandra Bakke cultivar nurturing farms of Nanjangud, Karnataka, INDIA (geographical coordinates: 12°9′12.88″ N, 76°42′51.01″ E). A plant taxonomist, Professor Siddaramaiah, authenticated a voucher specimen (MYS 457869), and the sample was deposited at the herbarium of the Department of Horticulture, Government of Karnataka, Mysore, India. The freshly uncut green jackfruits were washed under running water, followed by a thorough cleaning with sterile water, cut into slices and shade dried. Later, the dried pieces were ground into flour using a homogeniser. The whole jackfruit flour thus generated was stored at 4 °C till further analysis.

### 2.1. Extraction

The whole jackfruit flour was subjected to hot extraction with methanol using the Soxhlet apparatus. The extraction was carried out three times with 600 mL of methanol, and it was filtered after each extraction. After the filtrate was concentrated under vacuum using a rotary evaporator, the yield of the extract was assessed (Rotavapor R-200, Buchi, Switzerland) [9,10,11].

### 2.2. Analysis of Phenolic Compounds and Ascorbic Acid by HPLC

Identification of phenolic constituents from MJ was carried out as per the method described by Seal (2016) [12] with minor changes. In brief, the HPLC system (Agilent Technologies, Santa Clara, CA, USA) was operated at 37 °C with a flow rate of 0.8 mL/min and an injection volume of 20 µL, using a photodiode array (PDA) detector and a reverse phase C18 (250 mm 4.6 mm, Supelco) column. Methanol and 0.1% formic acid (*v*/*v*) formic acid in water (solvent A) made up the mobile phase (solvent B). The elution gradient was as follows: 85% A and 15% B from 0 to 55 min; 20% A and 80% B from 55 to 57 min; 85% A and 15% B from 57 to 60 min. The peak areas of the phenolic compounds were compared to those of the standards at 280 nm, which included gallic acid, chlorogenic acid, caffeic acid, vanillin, *p*-coumaric acid, quercetin, phydroxybenzoic acid, sinapic acid, and ascorbic acid.

### 2.3. In Vitro Enzymes Inhibition Assays

The inhibition studies for α-amylase (EC 3.2.1.1, categorised as type-VI B porcine pancreatic α-amylase), yeast α-glucosidase (EC 3.2.1.20, categorised as type-1 α-glucosidase), and human recombinant aldose reductase were carried out as previously described (Ramu et al. 2014) [13]. The IC_50_ values (µg/mL) were calculated from the least-squares regression line of the logarithmic concentrations plotted against % inhibition to represent the inhibitory activity of the tested compounds. This number represents the concentration of samples that can inhibit enzyme activity by 50% when compared with that of the control.

### 2.4. HSA Glycation Inhibition Assay at Multiple Stages

As per the protocol defined from previous work [13], the antiglycation potential at several stages was assessed, including early glycation product (fructosamine), intermediate (protein carbonyls), and late-stage glycation end products (AGEs). The HSA/fructose system was used to measure albumin glycation, which consisted of adding 1 mL HSA (10 mg/mL) to a 4 mL assay system containing 1 mL fructose (300 mM), 1 mL phosphate buffer (100 mM, pH 7.4 with 0.02 percent sodium azide), test samples, or aminoguanidine (as positive control) at various concentrations. The reaction mixture was incubated for 21 days at 37 °C by maintaining sterile conditions. The fluorescence intensities of the AGE generated (fluorescent products) from albumin glycation and control were measured using a spectrum scanning multimode reader (Thermo Fisher Scientific, Rockford, IL, USA) at excitation and emission wavelengths of 370 and 440 nm (with slit = 10 nm). The carbonyl group was measured as a marker for protein oxidative damage, while the NBT test was utilised to detect fructosamine (an Amadori product) synthesis in the glycated HSA and the control [13]. The percent inhibition for test samples was computed as specified in enzyme inhibition section.

### 2.5. Antioxidant Activity

The antioxidant capacity of MJ was evaluated by the commonly used in vitro test methods of DPPH, superoxide, and ABTS radical scavenging activities as previously described [14]. All the tests were performed in triplicates, with butylated hydroxyl anisole (BHA) as the positive control. EC_50_ values were used to represent their potential radical scavenging capability. An EC_50_ value implies scavenging of 50% of free, cation, and anion radicals.

### 2.6. Isolation and Identification of Bioactive Compounds from MJ

The bioactive compounds were extracted from 500 g of green jackfruit flour using solvents of increasing polarity, as reported by Ramu et al. 2014 [13]. The samples were tested for DPPH, ABTS and superoxide radical-scavenging activities after removing water and solvent by lyophilisation and flash evaporation. Because of its high yield and antioxidant properties, methanol extract was chosen for the extraction of bioactive components. On a silica gel (100–200 mesh) column (length 90 cm and diameter 3 cm) chromatography (elution rate of 2 mL/min flow with a total elution of 500 mL), 70 g of methanol extract was eluted stepwise with a linear gradient of chloroform, ethyl acetate, *n*-butanol, and methanol. Flash evaporation was performed for all the fractions collected. Similar fractions were pooled and concentrated after TLC analysis and then tested for antioxidant activities. Fractions were collected and spotted on pre-coated silica gel F254 plates (20 × 20 cm, Merck, Darmstadt, Germany). The plates were sprayed with NP/PEG (Natural products-polyethylene glycol reagent) to observe the spots, and the optimum resolution was reached in the chloroform: ethyl acetate: acetone: formic acid (4:3:2:1 *v*/*v*) solvent system. In the TLC pattern, the ethyl acetate (Fr. I; 8–16) fraction exhibited an equivalent retention factor (Rf). As a result, the fractions were grouped together and concentrated. The resultant concentrate was re-chromatographed and eluted progressively using chloroform: methanol linear gradients (100:0, 99:1, 98:2, 97:3, 97.5:2.5, 95:5, 92.5:7.5, 90:10 *v*/*v*) (Appendix A). The separation of caffeic acid (CA) and syringic acid (SA) from Fr. II (71–79) and Fr. II (97–101) fractions, respectively, was achieved by a single spot-on TLC in a suitable solvent solution (Figure 1). This answered Ferric chloride test for phenolic acids.

### 2.7. Spectral Measurements

On a Bruker DRX-400 spectrometer (Bruker Biospin Co., Karlsruhe, Germany), the 1H and 13C NMR spectra were recorded in CDCl3 with tetramethylsilane (TMS) as an internal standard. Chemical shifts in relation to the TMS signal were measured in parts per million (δ), and coupling constants were supplied in Hz. In ESI mode, the mass spectrum was obtained using an LCMS2010A (Shimadzu, Tokyo, Japan) with probes APCI and ESI. The IR spectra were collected using KBr discs in the 400 to 4000 nm range on a NICOLET 380 FT IR spectrometer (Thermo Fisher Scientific, Rockford, IL, USA). The ultraviolet (UV) spectra of the chemicals in methanol were captured using a Shimadzu UV-1800 spectrophotometer. Using an electrically heated VMP-III melting point equipment, the melting points were calculated without correction. Further elemental analysis of the compounds was performed using a Perkin Elmer 2400 elemental analyser.

### 2.8. α-Glucosidase, α-Amylase and Aldose Reductase Inhibition Kinetics

The activity of α-glucosidase, α-amylase and aldose reductase was measured in the presence or absence of the inhibitor (isolated compounds: caffeic acid and syringic acid) at various doses (IC_20_, IC_40_and IC_60_) of 4-nitrophynyl-d-glucopyranoside, starch and DL-glyceraldehyde as substrates, respectively. The Lineweaver-Burk (LB) plot was constructed by plotting 1/enzyme activity (1/v) versus 1/substrate (1/[S]) [15]. For CA and SA, the LB plot was used to determine the mode of inhibition, V_max_ and K_m_ value. Secondary plot (Appendix A) was used to establish the inhibitory constant (Ki) [16].

### 2.9. Molecular Docking Simulation

Protein and ligand preparation was performed according to the previous study conducted by Patil et al. (2021a) [17]. The protein sequence of *Saccharomyces cerevisiae* α-glucosidase MAL-32 (UniProt ID: P38158) obtained from UniProt was used to construct a homology model using SWISS-MODEL. The model was built with the help of X-ray crystal structure of *S. cerevisiae* isomaltase (PDB ID: 3AXH), showing 72% identical and 84% similar sequence at a resolution of 1.8 Å [18]. In order to validate the homology-built model of α-glucosidase, various tools were used. The PROCHECK tool was employed [19] to understand the stereo-chemical quality of the modelled protein structure. The ERRAT [20] and PROVE [21] tools were used to assess the overall quality of the protein model in terms of its reliability and stability. In addition, physicochemical properties were calculated using Expasy-ProtParam tool [22].

The X-ray crystal protein structures of α-amylase (PDB ID: 1DHK) and human aldose reductase (HAR) (PDB ID: 1IEI) were retrieved from RCSB PDB database. For the protein preparation for docking simulation, AutoDock Tools 1.5.6 software was used [23]. The binding site prediction of the protein molecules was performed based on the literature analysis. In case of α-glucosidase, the binding residues were placed in a grid box measuring 30 Å × 30 Å × 30 Å positioned at the coordinates x = −17.489 Å, y = −8.621 Å, and z = −19.658 Å using AutoDock Tools 1.5.6 [17]. Similarly, for α-amylase (22.48 Å × 22.48 Å × 22.48 Å positioned at x = 103.469 Å, y = 37.176 Å, z = 19.607 Å), and HAR (8.25 Å × 8.25 Å × 8.25 Å positioned at x = −5.06 Å, y = 0.19 Å, and z = 9.94 Å) the grid boxes consisting of binding residues were placed using AutoDock Tools 1.5.6. Ligand structures of caffeic acid, syringic acid, acarbose, quercetin, and aminoguanidine were retrieved from PubChem database. After ligand preparation, the compounds were docked into their respective protein targets using AutoDock Vina 1.1.2. Acarbose was considered as a control for both α-glucosidase and α-amylase, whereas quercetin was used as a control for HAR.

### 2.10. Molecular Dynamics Simulation

For the molecular dynamics simulation, the docked conformations of protein and respective ligands with most negative binding affinity were selected, and the simulation was performed according to the study conducted by Patil et al. (2021b) [24]. For the simulation, the GROMACS-2018.1 biomolecular software suite was utilised [25]. The system consisted of Ubuntu desktop workstation with Intel^®^ Core™ i7-11700 x64-based CPU, NVIDIA GeForce RTX 3060 Ti GPU (8 GB GDDR6), and 32 GB (16 × 2) DDR4 RAM. The CHARMM36 force field was used to assign all of the protein–ligand complexes, and the CGenFF server was used to acquire the ligand topology [26]. The pdb2gmx module of the GROMACS was used to add hydrogen atoms to the heavy atoms present. After that, the steepest descent technique was used to complete 5000 steps of vacuum minimisation. All of the protein–ligand complexes were arranged in a box with a 10-foot radius around the edges. The solvent was incorporated into the TIP3P water model. By introducing the right amount of Na^+^ and Cl^−^ counter ions, the entire system was neutralised. A total of 9 protein–ligand complexes were prepared which included the following—α-glucosidase-acarbose: 9472 residues, α-glucosidase-caffeic acid: 9430 residues, α-glucosidase-syringic acid: 9430 residues, α-amylase-caffeic acid: 7601 residues, α-amylase-syringic acid: 7601 residues, α-amylase-acarbose: 7643 residues, HAR-caffeic acid: 5078 resides, HAR- syringic acid: 5078 resides, and HAR-quercetin: 5089 residues. Along with these, 3 protein backbone atoms (bare protein) were also prepared for simulation. Using the steepest descent and conjugate gradient approaches, the energy of the resulting systems was reduced. It was then followed by a brief (1000 ps) equilibration in the NVT ensemble and then an NPT ensemble (1000 ps). At 310 K temperature and 1 bar pressure, all simulations took 100 ns. All the simulations were performed in triplicates. Using the XMGRACE software, a trajectory analysis of root-mean-square deviation (RMSD), root-mean-square fluctuation (RMSF), radius of gyration (Rg), and solvent accessible-surface-area (SASA) parameters was performed, with the findings shown in graphical representation [27]. Furthermore, hydrogen bond mapping has been performed for the residues bound with hydrogen bonds to analyse the catalytic mechanism of the experimental molecules, using the md distance utility of GROMACS software.

### 2.11. Druglikeness and Pharmacokinetics Analysis

The chemical structures of the experimental compounds were submitted to the ADMETlab 2.0 server in SMILES format for druglikeness and pharmacokinetic analyses. Lipinski’s rule of five was applied to the druglikeness assessment. In pharmacokinetic studies, characteristics such as MDCK permeability, Caco-2 cell permeability, volume distribution (VD), BBB (blood–brain barrier), cytochrome P (CYP) inhibition, clearance (CL), human Ether-à-go-go-related Gene (hERG), and the AMES carcinogenicity test were taken into consideration [28].

### 2.12. Binding Free Energy Calculations

The outcomes of the molecular dynamics simulation were used to calculate the binding free energies of the protein–ligand complexes using the Molecular Mechanics/Poisson-Boltzmann Surface Area (MM-PBSA) technique, according to the previous study conducted by the authors Patil et al. (2021b) [24]. It is another application of molecular dynamics simulations and thermodynamics for determining the extent of ligand binding with protein. The g_mmpbsa programme with MmPbSaStat.py script, which utilises the GROMACS 2018.1 trajectories as input, was used to determine the binding free energy for each ligand-protein combination. In the g_mmpbsa programme, three components are used to calculate the binding free energy: molecular mechanical energy, polar and apolar solvation energies. The calculation is performed using molecular dynamics simulation trajectories of last 50 ns were considered for computing ΔG with dt 1000 frames. It is evaluated using molecular mechanical energy, polar and apolar solvation energies.

### 2.13. Statistical Analysis

All the analyses were carried out in triplets. The data were provided as a mean with a standard deviation. To compare the treatment groups to the control group, one-way analysis of variance (ANOVA) was employed, followed by Duncan’s multiple range test using SPSS Software (version 21.0, Chicago, IL, USA). The results were considered statistically significant if the ‘*p*’ values were 0.05 or lower. GraphPad PRISM was used to calculate the IC_50_/EC_50_ values (version 4.03).

## 3. Results

In the present study, we established the antihyperglycaemic potential of methanol extract of jackfruit flour. In this regard, the isolation of caffeic acid and syringic acid was achieved after repeated chromatographic separations using silica gel column chromatography (Figure 1). The compounds’ structural elucidation was accomplished using a variety of physicochemical and spectroscopic techniques (UV, IR, ^1^H NMR, ^13^C NMR and MS). The following are the details of their structures: caffeic acid was recovered as a pale yellowish semi-solid substance from Fr. II (71–79). m.p. 223.5 °C. UV (methanol): λ_max_ 327 nm. IR (KBr): 1622 (Aryl-substituted C=C), 1730 (−COOH), 3510–3600 cm^−1^ (OH). ^1^H NMR (DMSO): δ 4.5 (bs, 2H, 2-OH), 6.3 (dd, *J* = 7 Hz, CO–CH=C), 6.5 (s, 1H, Ar–H), 6.8 (dd, 2H, Ar–H), 10.9 (s, 1H, −COOH). ^13^C NMR (DMSO): δ 113.5, 115, 117, 120, 127, 145, 147, 149, 171. MS: *m*/*z* 181 (M + 1). Analytical calculated data for C_9_H_8_O_4_ (180.04): C, 60.00; H, 4.48. Found: C, 60.04; H, 4.45%.

Syringic acid was obtained from Fr. II (97–101) as a yellowish semi-solid substance. m.p. 205.5 °C. UV (methanol): λ_max_ 217 nm. IR (KBr): 1745 (−COOH), 3550–3600 cm^−1^ (OH). ^1^H NMR (DMSO): δ 3.5 (s, 6H, O–CH_3_), 4.3 (bs, 1H, −OH), 7.2 (s, 1H, Ar–H), 7.3 (s, 1H, Ar–H), 10.6 (s, 1H, −COOH). ^13^C NMR (DMSO): δ 56 (2), 108 (2), 124, 138, 152 (2) 168. MS: *m*/*z* 199 (M + 1). Analytical calculated data for C_9_H_10_O_5_ (198.05): C, 54.55; H, 5.09. Found: C, 54.57; H, 5.05%.

Compounds (CA and SA) were identified as caffeic acid and syringic acid based on the above-mentioned results and a comparison with NMR and MS data in the literature [25,26,27]. These findings support our HPLC findings that CA and SA were present as components in MJ (Table 1). Furthermore, the total phenolic content of CA (81.33 mg GAE per g) and SA (83.40 mg GAE per g) was found to be high.

### 3.1. In Vitro Inhibition of Carbohydrate Hydrolysing Enzymes

In the present study, MJ and its isolated compounds (CA and SA) exhibited effective inhibitory potential in vitro against α-glucosidase. The IC_50_ values for MJ, CA, and SA were found to be 10.0, 8.0, and 7.50 µg/mL, respectively. On the other hand, acarbose (therapeutic drug) exhibited an IC_50_ value of 11.0 µg/mL, indicating that MJ, CA, and SA exhibited greater inhibitory potential in this experiment. With respect to IC_50_ values, it is clear that CA and SA inhibited yeast α-glucosidase strongly and were significantly (*p* ≤ 0.05) greater than acarbose and MJ (Table 2). The order of inhibitory effect was SA > CA > EaFr. > MJ > Acarbose.

Similar experiments were conducted on MJ and its derivatives against α-amylase, another major carbohydrate hydrolysing enzyme. The 50 percent inhibition of α-amylase by MJ and its active components is shown in Table 1. SA (IC_50_: 25.25 µg/mL) was shown to have the strongest inhibitory activity when compared to CA (IC_50_: 26.90 µg/mL), whereas MJ had the lowest inhibitory activity (IC_50_: 26.90 µg/mL). Overall, MJ, CA, and SA had a lesser (*p* ≤ 0.05) α-amylase inhibitory effect (based on IC_50_ values) than the positive control acarbose (IC_50_: 28.0 µg/mL). Information activity of every fraction has been given in the Appendix A.

### 3.2. Kinetics of α-Glucosidase and α-Amylase Inhibition

Because of their substantial inhibitory impact, CA and SA were chosen for additional kinetic inhibition experiments against α-glucosidase and α-amylase. Lineweaver Burk plots showed that the crossing point for different concentrations of CA (Figure 2A,C) and SA (Figure 2B,D) comes from the same y-intercept as the uncontrolled enzyme, despite the different slopes and x intercepts. The slope and the vertical axis intercept arose when CA and SA concentrations increased, but the horizontal axis intercept (−1/Km) increased as well. The kinetic data revealed that the maximum velocity (Vmax) of CA and SA reactions catalysed by α-glucosidase and α-amylase remained constant (with increasing concentrations) (Table 3). These findings suggested that the mechanism of α-glucosidase and α-amylase inhibition for both the compounds (CA and SA) was reversible and that it followed the conventional pattern of competitive inhibition. The inhibitory constant (Ki) for α-glucosidase and α-amylase was 1.03 and 0.52 mg for CA and 1.25 and 0.96 mg for SA, respectively, as measured by secondary plots (Table 3).

### 3.3. In Vitro Inhibition of Aldose Reductase

EF and its isolated compounds were identified to have a stronger inhibitory effect on the aldose reductase enzyme than quercetin, a phenolic inhibitor. With an IC_50_ value of 3.75 µg/mL, MJ inhibited aldose reductase (Table 2). CA (IC_50_: 3.10 µg/mL) isolated from MJ was found to be a strong inhibitor; however, it showed (in terms of IC_50_ values) a weaker inhibitory effect than SA (IC_50_: 3.0 µg/mL). MJ and its components were found to be potent inhibitors (*p* ≤ 0.05) than quercetin (IC_50_: 4.10 µg/mL).

### 3.4. CA and SA Inhibition of Aldose Reductase: A Kinetic Study

Other than the varied slopes and y-intercepts, LB plots demonstrated that the intersecting point for diverse concentrations of CA (Figure 2E) and SA (Figure 2F) emerges from the same x-intercept as an uncontrolled enzyme. The slope and vertical axis intercept both arise as the concentrations of CA and SA increased, whereas the horizontal axis intercept (1/Km) remained constant (Table 3). The kinetic data showed that CA and SA slowed the maximum velocity (Vmax) of the aldose reductase catalysed process (with increasing concentrations) without affecting the Km values (Table 3). These findings suggested that the mechanism of aldose reductase inhibition was reversible, similar to the non-competitive inhibition pattern. The inhibition constant (Ki) for aldose reductase was 1.11 and 1.64 mg of CA and SA, respectively, as measured by secondary plots (Table 3).

### 3.5. Antioxidant Ability, TPC and TFC

A combination of in vitro experiments, including DPPH, ABTS, and Superoxide, were conducted to evaluate the capacity of the extract, fraction, and isolated compounds to scavenge free radicals, with BHA used as a positive control. The results, reported as EC_50_ values (mg of tests per mL) in Table 4, demonstrated that the compounds had higher radical scavenging capabilities (*p* ≤ 0.05) than the standard (positive control). CA and SA were more effective than crude extract and fractions in all of the tests employed in this investigation, with activity ascending in the sequence SA > SA > EaFr. > MJ > BHA. The results showed that the extract, ethyl acetate fraction, CA and SA had high antioxidant properties and were significantly higher (*p* ≤ 0.05) than the positive control (Table 4).

With an extraction yield of 143.07 g/kg, the methanol extract of green jackfruit flour was found to contain a significant flavonoid (601.05 mg QE/g) and phenolic (252.07 mg GAE/g) content. The findings indicated that the total flavonoids and phenols were concentrated in the MJ and that the compounds separated from these active extracts had high TFC and TPC (Table 4**)**.

### 3.6. Identification of Various Phenolic Acids, Flavonoids and Ascorbic Acid in MJ by HPLC

Flavonoids, sterols, tannins, alkaloids, coumarins, and phenolic substances were found in the early phytochemical analysis of MJ. HPLC analysis was also performed to obtain a detailed polyphenolic composition of MJ, and the results are shown in Table 1, indicating the presence of various phenolic acids, flavonoids including ascorbic acid.

### 3.7. In Vitro Inhibition of HSA Glycation at Multiple Stages

The inhibitory effects of various dosages of MJ, CA, SA, and aminoguanidine (25, 50, and 100 g/mL) on early glycation products (fructosamine), intermediate (protein carbonyls), and late-stage glycation moieties (AGEs) after 3 weeks of incubation are shown in Figure 3A–C. At different stages, incubation with MJ and its components inhibited albumin glycation in a dose-dependent manner. At the conclusion, it was clear that both CA and SA had stronger inhibitory actions (at all levels) than MJ at various doses. The generation of fructosamines was found to be inhibited by MJ and its components in the range of 78 to 84% (Figure 3A), while the formation of protein carbonyl compounds was found to be suppressed in the range of 78–89% (Figure 3B), and it was maximum decreased in the presence of SA. According to fluorescence experiments on AGEs (Figure 3C), MJ, CA, and SA can inhibit them in the range of 64–75%. In general, 21-day incubation for MJ, CA, and SA showed a stronger inhibition than the known inhibitor aminoguanidine at various doses.

The quantity of fructosamine and protein carbonyl content in the various incubated test samples (MJ, CA and SA) with aminoguanidine at 1, 2, and 3 weeks is given in Table 5. In comparison to MJ, CA, SA, and aminoguanidine samples at a concentration of 100 g/mL, the amount of fructosamine increased in the samples incubated with fructose and HSA (HSA/fructose) in all three weeks. The production of fructosamine was inhibited the most in SA, followed by CA and MJ. Aminoguanidine exhibited inhibition as well, but at a lesser level (*p* ≤ 0.05) in comparison to MJ and its constituents. Similarly, when fructose was added to HSA, the amount of protein carbonyl formation increased as compared to HSA alone. When compared to HSA incubated with fructose, the reduction of protein carbonyl production by MJ and its components was significantly stronger (*p* ≤ 0.05) than aminoguanidine.

In the present study, the potential of MJ and its components on thiol group modification was also explored. Treatment with MJ and its isolated compounds exhibited effective denaturation protection, resulting in a significant increase in the thiol levels. Table 5 shows the amount of available free thiol groups as well as HSA oxidative modification of MJ and its components. Both the isolated compounds and MJ demonstrated substantial protection against thiol oxidation in a dose-dependent manner, as seen in Figure 3D. In the presence of MJ, CA, and SA, the results indicated that thiol shielding values varied from 74–83% at a concentration of 100 µg/mL. Under the same test conditions, aminoguanidine at 100 µg/mL exhibited 71 percent thiol group protection.

### 3.8. Molecular Docking Simulation

The modelled structure of α-glucosidase was validated using PROCHECK tool to understand the stereo-chemical quality of the modelled protein structure, which evaluate the backbone of phi-psi (Φ–Ψ) dihedral angles. The Ramachandran plot analysis showed that 88.1% of the residues were present in the favoured region, 11.1% of the residues were in the additional allowed region, 0.6% of the residues were in the generously allowed region, and 0.2% of the residues were included in the disallowed region. Results from the plot indicated that the proposed model is reliable. Further, to evaluate the statistics of non-bounded interaction between atoms ERRAT tool was used, which shows that the overall quality of the model, such as its reliability and stability, is 94.912. To evaluate the quality of the protein structure, PROVE tool was used, which evaluates the Z-score root-mean-square deviation (Z-score rms) to measure the average magnitude of irregularities. The scored atom value was found to be 3.0% of the standard deviation, which is considered away from the mean of that atom type. The total number of buried outlier protein values is 77. Results from the protein validation are given in Figure 4.

In addition, the physiochemical properties were calculated using Expasy-ProtParam server tool. The protein had 101,724 numbers of amino acids, with an estimated molecular weight of 12,290,969.97 Da and a theoretical pI of 6.63. The instability index was calculated was 37.91, which showed that the protein was stable. The hydrophilicity was calculated by the grand average of the hydrophilicity (GRAVY) of −0.044 and the aliphatic index as 42.79.

The compounds identified in HPLC were subjected to molecular docking simulation and were virtually screened for the best compound using the parameters such as binding affinity, the total number of non-bonding interactions, and the total number of hydrogen bonds. The summary of virtual screening has been given in Table 6. During the molecular docking of α-glucosidase with the experimental compounds, caffeic acid was able to bind inside the binding pocket of the protein with an affinity of −8.2 kcal/mol. It was found to bind deep inside the binding pocket with a total of six non-bonding interactions, which included four hydrogen bonds with ASP 349 (2.76 Å), GLU 278 (2.48 Å), ASP 214 (6.81 Å), and ARG 312 (2.67 Å). In addition, it was also able to bind through two electrostatic π-anion bonds with ASP 349 (4.80 Å) and ARG 439 (4.29 Å). However, the binding of syringic acid was more effective in comparison with caffeic acid. Similarly, syringic acid is also bound deep inside the pocket to form 10 non-bonding interactions, including four hydrogen bonds. The hydrogen bonds included ARG 439 (2.74 Å), ASP 68 (3.80 Å), GLU 276 (2.68 Å), and PHE 157 (2.60 Å). Along with these, syringic acid also formed a hydrophobic π-π bond with PHE 177 (5.30 Å), two alkyl bonds with PHE 158 (5.39 Å) and PHE 177 (4.53 Å), and three π-alkyl bonds with ALA 278 (4.00 Å), PHE 300 (5.16 Å), and PHE 157 (4.74 Å). Due to the extensive hydrophobic bonds, the binding affinity of syringic acid was found to be −11.4 kcal/mol. However, acarbose was predicted with lower binding efficiency compared to syringic acid. With a binding affinity of −10.2 kcal/mol, it formed a total of seven non-bonding interactions. The six hydrogen bonds included PRO 309 (2.31 Å and 2.94 Å), HIS 239 (2.28 Å), ASN 241 (2.18 Å), ASP 408 (1.85 Å), and ARG 439 (2.40 Å). It also formed a hydrophobic π-σ bond with HIS 279 (3.38) and was predicted with two unfavourable bonds with THR 307 (2.99 Å) and ASP 349 (2.83 Å). Visualisation of docking results for α-glucosidase has been shown in Figure 5.

In the case of α-amylase, caffeic acid was able to bind to the key residues of the binding pocket located between loop one and loop two. It was bound to the catalytic residue ASP 197 with a hydrogen bond (2.37 Å) and one electrostatic π-cation bond (4.80 Å). It was also bound to another catalytic residue ASP 300 with an electrostatic π-anion bond (4.95 Å). However, it could not form any bond with the third catalytic residue, GLU 233. Furthermore, there were other hydrogen bonds with GLN 63 (2.75 Å), TYR 62 (2.68 Å), and a π-π hydrophobic bond with TYR 62 (4.69 Å). With these six non-bonding interactions and three hydrogen bonds, the binding affinity of caffeic was found to be −8.1 kcal/mol. In comparison with caffeic acid, syringic acid was predicted with extensive binding interactions and robust affinity. It formed a total of 11 non-bonding interactions with three hydrogen bonds. It bound all the three catalytic residues of the binding pocket through hydrogen bonds with GLU 233 (2.77 Å), ASP 300 (3.51 Å), and electrostatic π-cation bond with ASP 197 (4.30 Å). Apart from these, it bonded with TYR 62 (2.61 Å) with a hydrogen bond. Hydrophobic bonds dominated the interaction of syringic acid with α-amylase. It formed a π-π bond with TYR 62 (5.29 Å), three alkyl bonds with TRP 58 (4.87 Å), ALA 198 (5.38 Å), and HIS 201 (4.74 Å), and three π-alkyl bonds with LEU 162 (4.06 Å), ALA 198 (4.19 Å), and HIS 299 (4.41 Å). With these interactions, the binding affinity of syringic acid was found to be −12.5 kcal/mol. In the case of acarbose, the binding interaction and affinity were found to be inferior compared to that of caffeic acid and syringic acid. The total number of binding interactions were found to be 2, while both of them were hydrogen bonds with key residues GLU 233 (2.87 Å) and ASP 197 (3.40 Å). In addition, three unfavourable bonds with ASP 300 (2.82 Å), ASN 298 (2.49 Å), and ARG 195 (2.50 Å). The binding affinity of acarbose with α-amylase was found to be −6.2 kcal/mol. The binding interactions of experimental compounds with α-amylase have been given in Figure 6.

Upon binding with HAR, caffeic acid occupied the same binding site of the co-crystallized ligand zenarestat, which is located near the NADPH binding site. However, caffeic acid was able to form only four non-bonding interactions, with two hydrogen bonds. It was bound to only three key residues, namely LEU 300 (3.63 Å) with a π-σ bond, TRP 111 (5.15 Å) with a π-π bond, and TYR 309 (2.07 Å) with a hydrogen bond. It was also bound to CYS 80 (3.41 Å) with another hydrogen bond. The binding affinity of caffeic acid was found to be −7.4 kcal/mol. In the case of syringic acid, the binding was to five of the six key residues, including LEU 300 (4.43 Å) with a π-alkyl bond, HIS 110 (2.76 Å and 3.04 Å) with two hydrogen bonds, TYR 48 with a hydrogen bond (2.11 Å) and π-alkyl bond (5.48 Å), CYS 298 (3.72 Å) with a hydrogen bond, and TRP 111 (5.15 Å) with an alkyl bond. The other non-bonding interactions included TRP 219 (5.08 Å), VAL 47 (3.72 Å), TRP 20 (4.38 Å), with π-alkyl bonds, TRP 20 (4.50 Å) with a π-π bond, TRP 20 (2.60 Å) with a hydrogen bond. Due to the extensive interaction with residues through hydrophobic bonds, the binding affinity of syringic acid was found to be −12.9 kcal/mol. In the case of quercetin, the compound was bound to only three of the six key residues. It was bound to TRP 111 with hydrophobic π-π bonds (5.74 Å and 5.22 Å), LEU 300 with one π-alkyl bond (5.13) and one π-σ bond (3.63 Å), CYS 298 with a hydrogen bond (2.47 Å). The other residues were TRP 20 (5.03 Å) and VAL 47 (4.38 Å) with π-alkyl bonds. It also formed two unfavourable bonds with TYR 309 (2.42 Å) and CYS 298 (2.39 Å). In total, quercetin formed seven non-bonding interactions with one hydrogen bond, with a binding affinity of −10.3 kcal/mol. The visualization of binding interactions of the experimental molecules with HAR has been depicted in Figure 7.

### 3.9. Molecular Dynamics Simulation

Several parameters such as the protein–ligand complex RMSD, RMSF, Rg, SASA, ligand RMSD, and ligand-hydrogen bonds were studied during molecular dynamics simulation to assess the complex’s overall stability. The RMSD plot of the protein–ligand combination displays the ligand’s stability inside the binding pocket over the course of a 100 ns simulation. The RMSF of a protein–ligand complex, on the other hand, is used to compute the average deviation of a particle (e.g., a protein residue) over time from a reference site. As a result, RMSF focuses on the protein structural regions that are the most/least different from the mean. In addition, the radius of gyration (Rg) shows the structural compactness of the molecules by calculating the root-mean-square distances with respect to the central axis of rotation. The area around the hydrophobic core created between protein–ligand complexes were shown in SASA plots for all protein–ligand complexes. During the simulation, the greatest number of H-bonds remained consistent with the molecular docking, and only a few bonds were concurrently broken and rebuilt. Therefore, ligand-hydrogen bonds also play a crucial role in dynamic trajectory analysis.

In the case of α-glucosidase, the RMSD plot depicts that the protein backbone atoms became stable after 20 ns. However, all the protein–ligand complexes became stable after 15 ns protein–syringic acid complex and protein backbone atoms were equilibrated at 0.4 nm, while protein–acarbose and protein–caffeic acid complexes equilibrated at 0.25–0.30 nm. In Rg, the protein backbone atoms equilibrated at 3.1 nm. All the protein–ligand complexes were found to be equilibrated at 2.4 nm. In the case of RMSF, the protein model showed more fluctuations at C-terminal, N-terminal, and loop regions. The α-glucosidase protein molecule extends up to ~1000 residues. The protein–ligand complexes, however, are extended up to only ~600 residues. Residues from 600–900 showed minimal fluctuations that depict the stability of the protein backbone atoms. However, the C-terminal region of the α-glucosidase protein shows the highest fluctuation (~0.85 nm) at the residues 900–1000. However, in comparison with the other protein–ligand complexes, protein–syringic acid was found with lesser fluctuations. Similar to Rg, SASA of all the protein–ligand complexes were found to be similar, within the range of 225–250 nm^2^. Yet, the protein model was found with the SASA value of ~350 nm^2^. In addition, all three ligands: caffeic acid, syringic acid, and acarbose, were able to form three hydrogen bonds. Figure 8 describes the visualization of trajectories from dynamics simulation for α-glucosidase complexed with different experimental compounds.

During the dynamics simulation of α-amylase, the RMSD plot depicts that the protein model equilibrated at ~0.3 nm, with protein–syringic acid being equilibrated at 0.25–0.30 nm. However, both protein–acarbose and protein–caffeic acid complexes were equilibrated at 0.35–0.40 nm. In the case of Rg, all the protein–ligand complexes except caffeic acid were equilibrated at 2.3 nm, whereas the latter reached equilibration at 2.25 nm. In the RMSF analysis, protein–syringic acid complex was found with minimal fluctuations in comparison with the other complexes. The protein model showed maximal fluctuation only at the loop region at 350 residues. Apart from this, caffeic acid was found with maximal fluctuation at 240 residues at the other loop regions. In the case of SASA, all the protein–ligand complexes showed a similar pattern of equilibration, where the SASA value of all the molecules, including protein, was found to be ranging between 180–190 nm^2^. In the case of the ligand-hydrogen bonds, syringic acid formed more hydrogen bonds (*n* = 7). Figure 9 shows the visualization of trajectories from dynamics simulation for α-amylase complexed with different experimental compounds.

The trajectory analysis of HAR complexed with different experimental molecules shows that RMSD of the protein backbone atoms, protein–syringic acid, protein–quercetin complexes were equilibrated at ~0.35–0.40 nm. However, the protein–caffeic acid complex was found with 0.25 nm of RMSD. In the case of Rg, the protein backbone atoms and protein–syringic complex were found to be equilibrated at 1.850–1.875 nm. Yet, the other protein–ligand complexes were found with the Rg values of ~1.875–1.900 nm. During the RMSF plot analysis, it was found that the protein–caffeic acid has the highest fluctuations at the loop regions (100–150 residues) and C-terminal region. In comparison with the other complexes, protein–syringic acid complex was found with minimal fluctuations. However, in SASA analysis, all the plots, including protein backbone atoms, were found with a similar pattern of values, ranging between 130–140 nm^2^. The ligand hydrogen bonds of syringic acid were also found to be the highest of all the ligands simulated (*n* = 5). Figure 10 shows the visualization of trajectories from dynamics simulation for HAR complexed with different experimental compounds. Appendix A consists of the results of molecular dynamics simulations run in triplicates. In order to support the data of these simulations through the assessment of the catalytic mechanism of the ligands, hydrogen bond mapping was performed. Results of the hydrogen bond mapping have also been depicted in Appendix A.

### 3.10. Druglikeness and Pharmacokinetics Analysis

During the druglikeness evaluation, caffeic acid, syringic acid, aminoguanidine, and quercetin obeyed Lipinski’s rule of five. In the case of absorption, distribution, metabolism, excretion, and toxicity (ADMET) analysis, which is also known as pharmacokinetic analysis, only syringic acid was found with optimal Caco-2 cell permeability. In addition, quercetin was not able to clear the AMES mutagenicity test, thus revealing its probable carcinogenic effect. Therefore, druglikeness and pharmacokinetics reveal that syringic acid is the best potential drug candidate among the selected experimental compounds. Details of the druglikeness and pharmacokinetics assay have been given in Table 7.

### 3.11. Binding Free Energy Calculations

Binding free energy calculations indicate that all the protein–ligand complexes are majorly formed Van der Waal’s energy. This was followed by binding energy, electrostatic energy, and SASA energy. In comparison with the other compounds, syringic acid showed a higher binding free energy when complexed with the protein molecules. In this context, results from binding free energy calculations show that syringic acid is a stable inhibitor of all the target proteins, in accordance with the results from molecular dynamics simulation. Details of the binding free energy calculations have been depicted in Table 8.

## 4. Discussion

To optimize the frequency, progression, and severity of diabetic complications, successful hyperglycemia intervention is important. Using a single therapeutic strategy, on the other hand, has not been successful in preventing all of the adverse implications of high blood glucose levels. As a result, α-glucosidase inhibitors, aldose reductase inhibitors, antiglycation medicines, and antioxidants may be a potential option for reducing the adverse effects of glucose. Numerous studies have shown that crude plant extract, as well as bioactive combinations derived from it, can help lower blood glucose levels [29]. In owing to its purported health benefits, jackfruit consumption has increased in recent years. The pulp and seeds of the jackfruit are high in commercially important substances with possible physiological benefits [6]. Routine dietary supplementation with jackfruit has been found to protect against and even treat a range of illnesses, such as stomach ulcers and cardiovascular disease; it may even help to prevent and delay the spread of certain cancers. Side effects are quite seldom reported [7]. During the jackfruit season in Kerala, an assessment and evaluation of the nutritional and glycemic value of green jackfruit as a diabetes alternative to rice revealed a reduction in the use of antidiabetic medicine [30]. In a recent study, green jackfruit flour (30 g per day) was found to be helpful in decreasing HbA1c, FPG, and PPG levels in T2DM patients when compared to the placebo flour [31]. To the best of our knowledge, no research has been performed on the therapeutic value of the whole unripe jackfruit in the treatment of T2DM. In the present study, the inhibitory effect of whole green jackfruit flour on carbohydrate hydrolysing enzymes (α-amylase, α-glucosidase), aldose reductase, and protein glycation were investigated in vitro using in vitro models.

In the conversion of dietary carbohydrates to glucose, the enzymes α-amylase and α-glucosidase are involved. Inhibition of these enzymes is being used to treat diabetic individuals since it delays carbohydrate breakdown and, as a result, slows down the glucose absorption in the intestine [32]. In comparison to the standard drug acarbose, the whole jackfruit flour extracted with methanol showed significant inhibition against α-amylase and α-glucosidase in the current study. Furthermore, MJ was subjected to repeated silica gel column chromatography, which led to the extraction of two phenolic compounds that were analytically pure using thin-layer chromatography. These two compounds were identified as caffeic acid and syringic acid, which belong to the phenolic acids family, using a variety of bioanalytical procedures. CA and SA, both extracted from the whole jackfruit, were investigated for their ability to operate as antihyperglycaemic activity by targeting important carbohydrate metabolism enzymes in bio-evaluation studies.

Caffeic acid is renowned for its antioxidant, anti-inflammatory and anticarcinogenic properties [33], while Syringic acid is known for its antioxidant, anti-inflammation, antimicrobial, antidiabetic, anticancer, and protection of the heart, liver and brain/CNS properties [34]. Furthermore, these compounds have been shown to exhibit significant biological activity, implying that they are effective in the treatment of a variety of metabolic illnesses. However, no studies on its role in antihyperglycaemia have been published to date. CA and SA, both extracted from MJ, were found to be α-glucosidase inhibitors in our investigation, with SA being the most potent. In order to study the mechanism underlying this inhibition, an LB plot was produced from the kinetics data, which revealed a reversible, competitive pattern of inhibition with low Ki values. Acarbose, a known therapeutic drug, demonstrated a similar finding on competitive inhibition [35]. MJ and its constituents, on the other hand, were tested for their ability to inhibit another important carbohydrate hydrolyzing enzyme, α-amylase. Yet, our results are in agreement with the previous study that most of the phenolic acids inhibit both α-amylase and α-glucosidase [36]. The MJ and its compound’s inhibitory action against α-amylase were much lower than that of acarbose. On α-glucosidase and α-amylase, LB plot analysis revealed that CA and SA showed a competitive mode of inhibition. The majority of plant-derived polyphenols inhibited α-glucosidase and α-amylase competitively [37,38].

Aldose reductase is involved in the polyol pathway as the first rate-limiting enzyme responsible for the conversion of glucose to sorbitol. In normoglycemic conditions, the glycolytic pathway metabolises the majority of glucose; however, in hyperglycemic conditions, the aldose reductase-associated polyol pathway increases dramatically, resulting in sorbitol buildup in the cells due to its low membrane permeability [39]. Synthetic aldose reductase inhibitors (Zopolrestat, Epalrestat, Sorbinil, and others) have been designed to treat and prevent diabetes complications by reducing the hyperglycemia-induced polyol pathway. However, due to a number of side effects and ineffectiveness, they are no longer widely used. As a result of their lower toxicity, natural aldose reductase inhibitors are important in the treatment and prevention of diabetes complications [40]. In the present study, MJ and its compounds (CA and SA) were strong aldose reductase inhibitors in comparison to the reference compound quercetin. On aldose reductase, the LB plot revealed that CA and SA exhibited non-competitive inhibition. Previous research has shown that most polyphenols inhibit aldose reductase in a non-competitive manner [41].

Excess glucose in the bloodstream causes the glycation of different proteins, rendering them inactive. Fructose has recently been discovered in glycation, either directly or through activating the polyol pathway to promote fructose production from glucose. According to studies, fructose participates in glycation at a faster rate than glucose, implying greater damage. The nucleophilic addition reaction between the free amino group of proteins and the carbonyl group of the reducing sugar initiates the creation of a Schiff’s base, which results in irreversible Amadori products such as fructosamine. The fructosamines in Amadori products are transformed to a variety of carbonyl compounds in the second stage of glycation, including glyoxal, methylglyoxal, and deoxyglucosones. Carbonyl proteins are formed as a result of this process, which results in the loss of protein thiols, which are promising protein oxidation markers [13]. As the glycation process progresses, insoluble fluorescent products, known as advanced glycation end products (AGE), develop, which bind with glycated proteins and accumulate in cells. They obstruct normal protein action within cells, and they cause irregular cross-linking of the extracellular matrix, obstructing its normal function. Aside from the vascular difficulties that AGEs cause, they also cause the production of reactive oxygen species (ROS), which is linked to the majority of diabetes issues [42].

In this regard, our study was designed to assess if MJ, CA and SA could prevent each stage of glycation and the production of AGE. Under in vitro conditions, the first stage of protein glycation was achieved by exposing HSA to high fructose concentrations, which was expected to glycate HSA. As expected, in our study, the fructosamine levels were higher than the non-glycated HSA levels, and MJ, CA, and SA treatment greatly decreased this, outperforming the usual inhibitor aminoguanidine. Furthermore, the second stage of protein glycation was revealed by the increased protein carbonyl groups formed after HSA was exposed to a high fructose load and the correspondingly low levels of protein thiols. Treatment of MJ and compounds reduced this to about 75–85 percent, indicating that it has a protective role. Finally, the ability of the compounds to generate fluorescent products was used to assess AGE formation, with the results indicating a significant reduction in AGE formation. As a result, the possible function of MJ and the compounds in preventing each stage of protein glycation is affirmed, implying that they could be useful in controlling various complications linked with it. The polyphenolic compounds found in the whole jackfruit extract may be responsible for its anti-glycation effect [2]. These findings were in accordance with the previous studies on the anti-glycation properties of polyphenol compounds [2,13,39]. The anti-glycation experimental agent aminoguanidine was less effective than the jackfruit extract tested.

Aminoguanidine, a well-known AGE inhibitor, works by inhibiting the formation of carbonyl intermediates. Other inhibitors, which act as metal chelating agents or antioxidants, are also implicated. However, the long-term consequences of these compounds include a hepatotoxic effect, necessitating the development of safer alternatives. Furthermore, these molecules capture all undesirable free radicals in vivo. In order to avoid these disadvantages, it is beneficial to boost antioxidant defences in order to avoid protein glycation [43]. Various in vitro tests, such as DPPH, ABTS, and superoxide, were used to examine the radical scavenging capabilities in the present study. In all of the assays used in the study, our findings showed that MJ and its constituents had increased free radical scavenging activity, implying that they play a protective function against free radical-mediated damage. The investigations also demonstrated that MJ has a high total flavonoid and phenolic content, which is linked to the extracts’ radical scavenging activity. The antioxidant potential of numerous plant extracts is well-documented, and most investigations have revealed that methanolic extracts outperform all other solvent extracts [44,45]. The extracts with the highest quantities of phenolic compounds and flavonoids were found to be the most active antioxidants in all of the studies, protecting against oxidative damage produced by free radicals in diseases such as cancer, diabetes, asthma, dementia, Parkinson’s disease, and others [46]. Similarly, our findings show that MJ and its phenolic compounds have significant antioxidant activity, implying that they have a lot of medicinal potentials. MJ also had a high total phenolic (0.96) and total flavonoid content (0.95), which is associated with the extracts’ radical scavenging action. The TPC is a primary predictor of antioxidant strength in plant extracts, according to Wang and his colleagues (2010) [47], which is consistent with our findings. According to these data, the total polyphenolics in MJ showed stronger antioxidant activity. Earlier research on mulberry fruits by Natic et al. (2015) [48] found that high quantities of polyphenols were responsible for the considerable antioxidant–antiradical scavenging activity and superoxide radical scavenging capability found in mulberry fruits. Furthermore, our findings demonstrated a substantial link between TFC and radical scavenging activity, and similar results were observed by Metrouh et al. (2015) [49]. MJ was further subjected to HPLC analysis to characterise the bioactive components responsible for pharmacological action. According to earlier research, phenolics and flavonoids were shown to be abundant in methanol extract of whole jackfruit flour, which have a strong link to improved biological and pharmacological activity [12,50].

Molecular docking simulation is performed to know the interaction of ligands with the target proteins at a molecular level. It determines the extent of ligand interaction which in turn shows the protein inhibition/activation. In this study, we have selected two phenolic acids from jackfruit methanol extract for targeting the inhibition of α-glucosidase, α-amylase, and HAR. Results from the docking simulation reveal that caffeic acid and syringic acid had the highest binding affinity with all the target proteins. During the docking simulation of α-glucosidase, all the experimental compounds were bound deep inside the binding pocket. The docking was accurate, according to the previous study conducted by Patil et al. (2021a) [17]. Although they were found inside the binding pocket, the interaction of the syringic was found to be superior to caffeic acid and acarbose. Since the number of hydrophobic bonds (π-π, π-alkyl, and alkyl) is considered stronger in comparison with the hydrogen bonds [51], the binding affinity of syringic acid was high. The same trend was demonstrated by syringic acid in the case of α-amylase and HAR. With all the three proteins, the binding was with greater hydrophobic bonds compared to other non-bonding interactions.

The binding interaction analysis of the compounds with α-amylase revealed that syringic acid is bound to all the three catalytic residues of the protein (GLU 233, ASP 300, ASP 197), which are present in the hydrophobic binding pocket located between loop one and loop two. Conversely, both caffeic acid and acarbose were not able to bind to all three catalytic residues. Binding to these catalytic residues would effectively reduce the activity of α-amylase, as indicated in recent studies [52,53]. Therefore, syringic acid proves to be a more efficient inhibitor than caffeic acid and acarbose.

Analysis of binding interactions from HAR indicates that all the ligands occupy the exact binding site as the co-crystallized ligand zenarestat, which is present in the vicinity of the NADPH binding site. HAR catalyzes the NADPH-dependent conversion of glucose to sorbitol, the first step in the polyol pathway of glucose metabolism [54]. Thus, it is essential to inhibit this portion of the enzyme. Although caffeic acid and quercetin were bound to the same region, they could not form the bonds with the key residues that bound to zenarestat. Out of the six key residues (TYR 309, LEU 300, TRP 111, CYS 298, HIS 110, and TYR 48), syringic acid was able to bind to five of them, whereas the other ligands were not able to bind to all of them. Since the binding interaction of syringic acid is efficient, it is expected to reduce the enzyme activity without any interference [55].

Molecular dynamics simulation is performed to assess the overall stability of the protein–ligand complex kept in s defined environment for a definite amount of time. Simulating experimental molecules with their target proteins has given variable results in this study. In case of α-glucosidase, all the plots describe syringic acid as the most stable compound. The protein–syringic acid complex was never predicted with abnormal fluctuations in any of the plots (RMSD, RMSF, Rg, SASA, and ligand hydrogen bonds). However, plots of protein–caffeic acid and protein–acarbose complexes were not as concurrent as the protein–syringic acid complex with the protein backbone atoms. The same trend was observed in the simulation of all four target proteins. The concurrent plots of protein–ligand complex with the protein backbone atoms indicate stronger binding affinity during the simulation study [56,57]. In the case of α-amylase, protein–acarbose and protein–caffeic acid complexes were proven to be relatively unstable when compared with that of the protein–syringic acid. The instability of acarbose has been reported in previous studies. In addition, simulation plots of a protein–syringic acid complex was in accordance with the previous studies [17,27]. Furthermore, simulation outcomes from HAR complexed with the experimental compounds depict the relative instability of the protein–caffeic acid. In this, the stability of protein–quercetin complex was on par with that of the protein–syringic acid complex. The results obtained with the simulation of HAR were in accordance with a study, which proposed kusunokinin as a potential inhibitor of HAR [58].

Druglikeness and pharmacokinetics analysis is performed to evaluate the bioavailability of the potential drug candidates. None of the above-mentioned studies has performed this assay to evaluate the bioavailability of their reported lead compounds. Yet, in our study, syringic acid was reported with zero risk of violating the druglikeness and pharmacokinetic parameters. It had passed the druglikeness test by obeying Lipinski’s rule of five, which is dependent on the physico-chemical parameters of the compound [59]. In addition, the compound was able to clear the pharmacokinetic parameters, which have been categorised, namely, absorption, distribution, metabolism, excretion, and toxicity. Except for syringic acid, all the compounds failed to pass the Caco-2 cell permeability test, indicating their ability to become absorbed. Caco-2 is a model of medication absorption in the human intestine. This model can be used to determine whether a molecule is acceptable for oral administration, predict intestinal permeability, and explore drug efflux [60]. Even though the docking and dynamics simulations were good, quercetin was found to be violating the AMES mutagenicity test, resulting from being carcinogenic [61] (Appendix A). Therefore, the druglikeness and pharmacokinetic analysis also reveal that syringic acid is the most favourable potential drug candidate.

## 5. Conclusions

This study is the first to investigate the antihyperglycaemic potential of the whole unripe jackfruit, as well as the identification of its polyphenol constituents. Moreover, MJ exerts a significant antihyperglycaemic effect by inhibiting carbohydrate hydrolysing enzymes, as well as reducing diabetes-related comorbidities by suppressing aldose reductase and AGE-related pathways. The beneficial effects of MJ accord with the positive effects of the isolated CA and SA, implying that they could be used as antidiabetic drugs. Our findings clearly establish MJ and its components for its antioxidant activities, implying a relationship with their corresponding antidiabetic effects. Furthermore, the identification of significant amounts of caffeic acid (0.52%) and syringic acid (0.58%) in MJ gave evidence for a strong chemical basis for MJ-associated antidiabetic capabilities. The hallmark outcome of this study is the identification of syringic acid as a lone potential inhibitor of all the selected targets of diabetes mellitus, which have their roles in different stages of vicious metabolic disorder. The phytocompound has surpassed all the other experimental compounds in the in silico investigations to become a common inhibitor of all the target proteins. In this context, we show that our study has identified a single, novel multi-target inhibitor for different proteins that act as targets in the stages of diabetes mellitus by playing a crucial role in the decadence of the metabolic state.

## Figures and Tables

**Figure 1 molecules-27-01888-f001:**
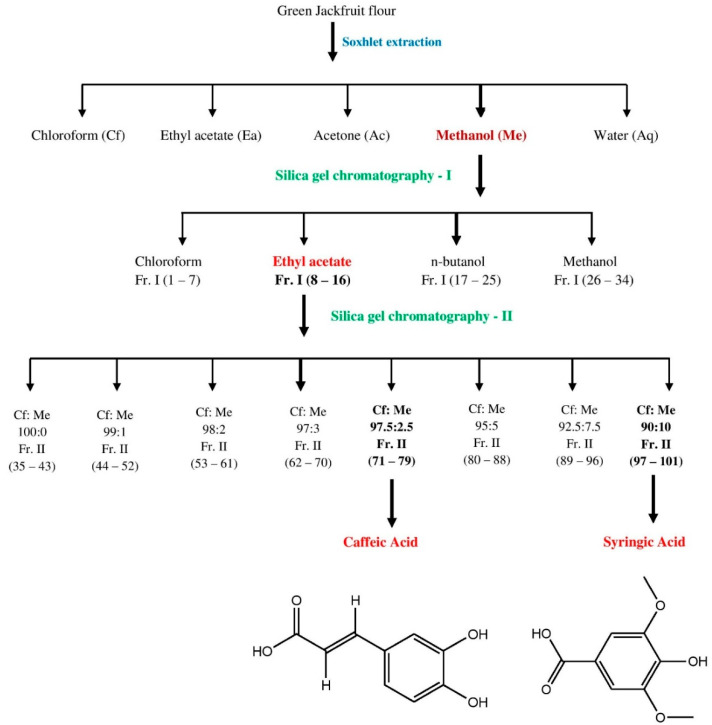
Purification of caffeic acid and syringic acid from whole jackfruit flour employing multiple solvent systems and silica gel chromatography.

**Figure 2 molecules-27-01888-f002:**
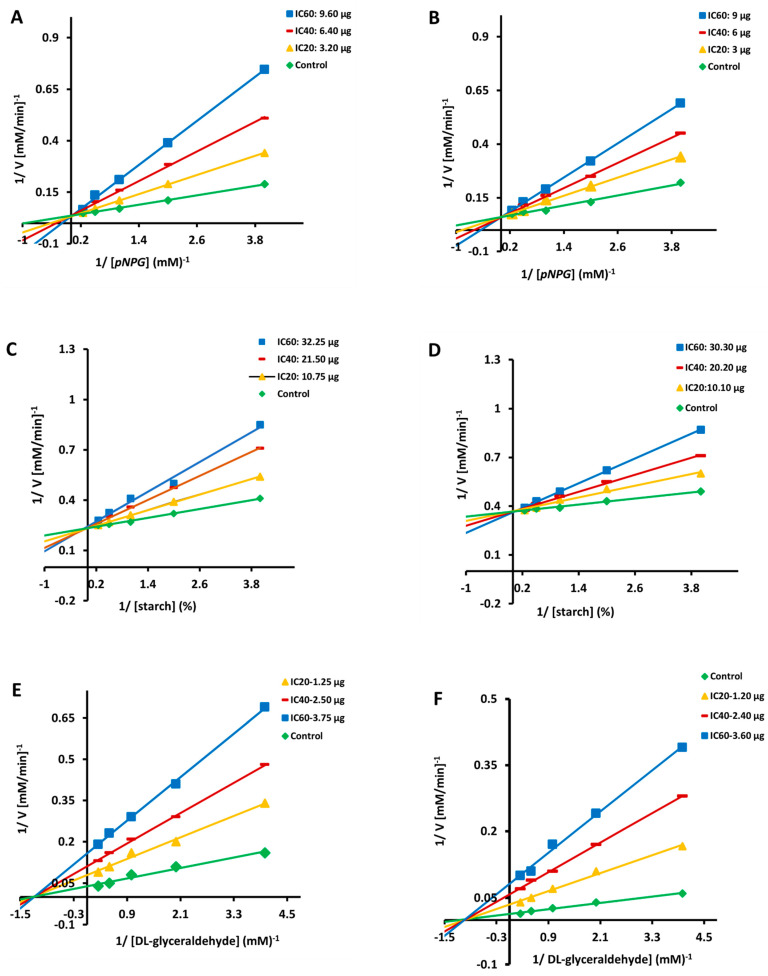
Double reciprocal plot of substrate dependent enzyme kinetics against α-glucosidase (**A**,**B**), α-amylase (**C**,**D**) and aldose reductase (**E**,**F**) inhibition by caffeic acid and syringic acid.

**Figure 3 molecules-27-01888-f003:**
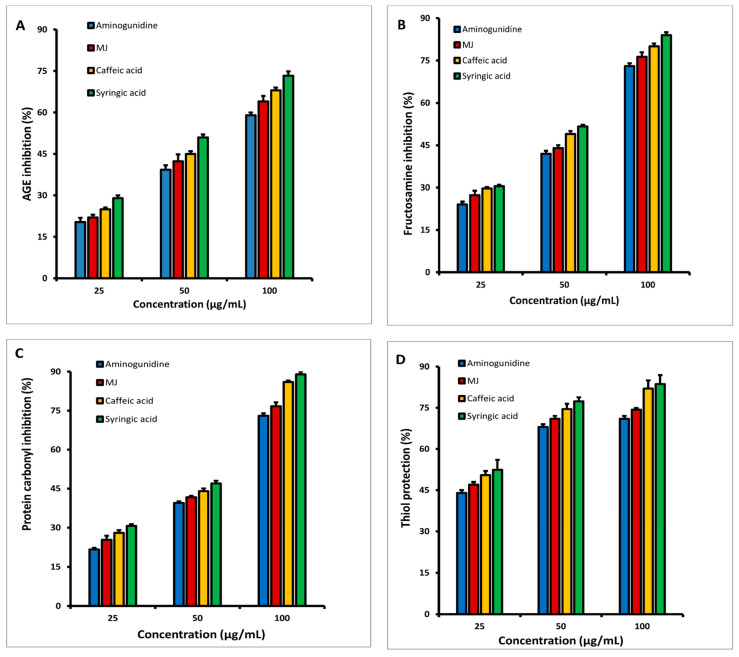
At various doses, the inhibitory effects of methanol extract of jackfruit flour (MJ) and its separated components on (**A**) Fructosamine, (**B**) Protein carbonyls, (**C**) AGE formation, and (**D**) Protein thiol protection.

**Figure 4 molecules-27-01888-f004:**
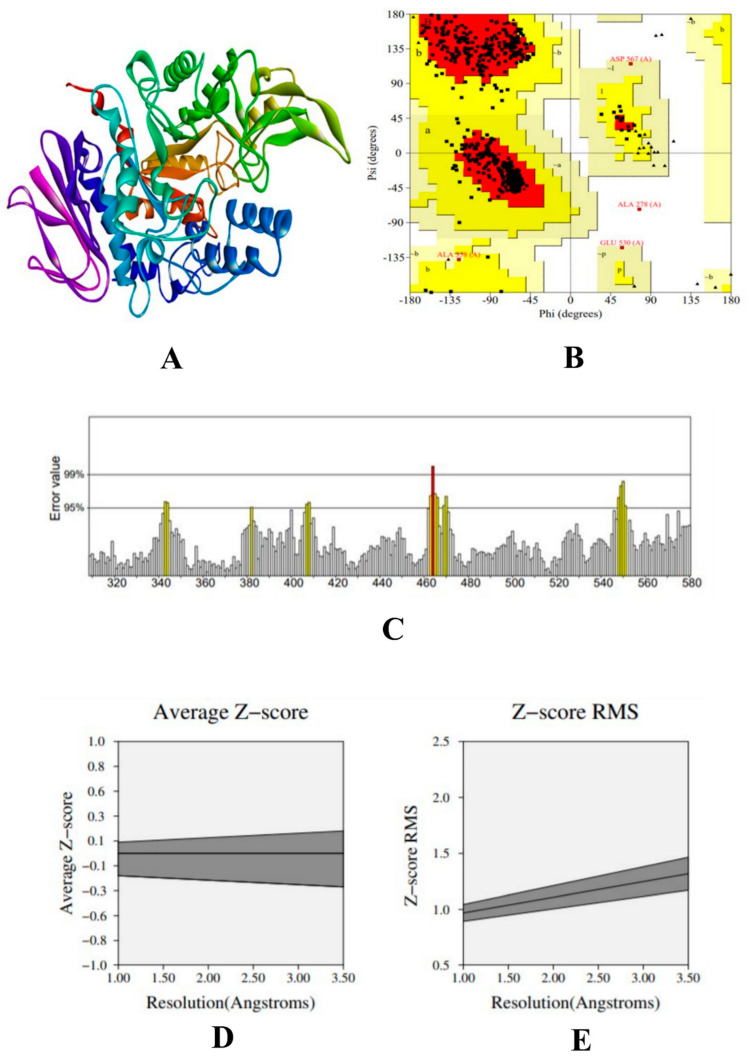
(**A**) 3D structure of homology-built model of α-glucosidase, (**B**) Ramachandran plot of the α-glucosidase model showing the residues present in the favoured region, (**C**) ERRAT result of the α-glucosidase model showing the overall stability and reliability, where the red line depicts the rejected part of the protein, (**D**) Z-score result the α-glucosidase model showing the average Z-score, and (**E**) Z-score RMS of the α-glucosidase model.

**Figure 5 molecules-27-01888-f005:**
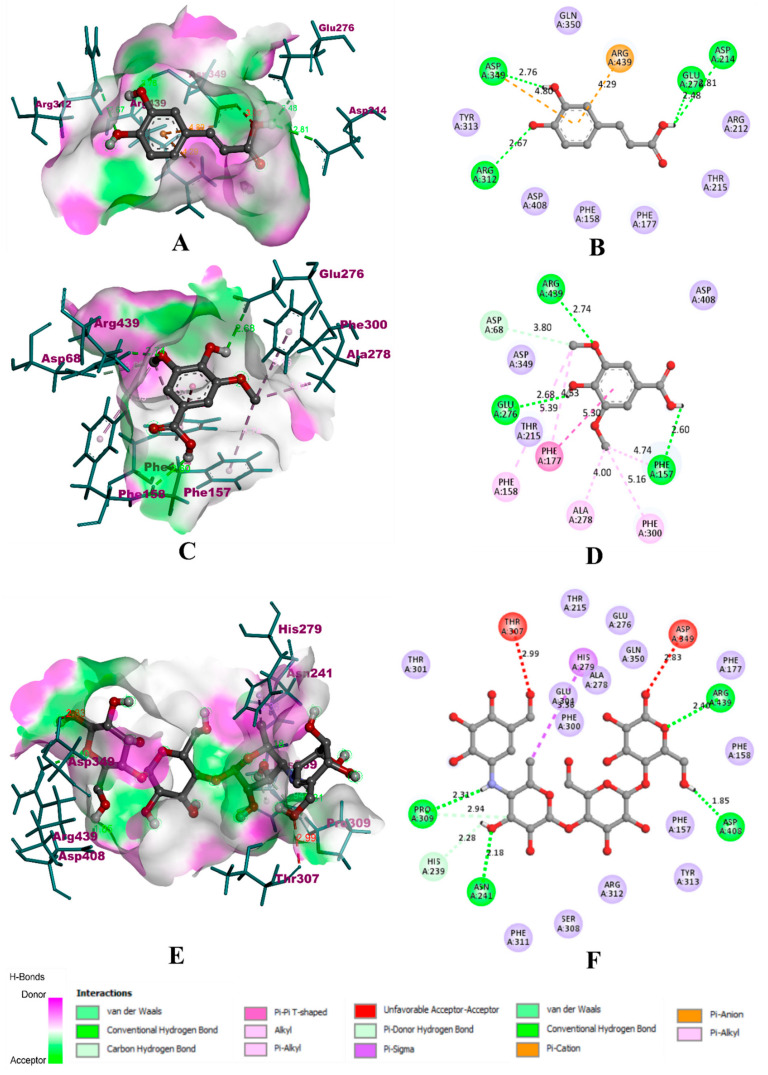
Visualization of docking simulation of experimental compounds with α-glucosidase. (**A**,**B**) Interaction of caffeic acid visualized in 3D and 2D, (**C**,**D**) Interaction of syringic acid visualized in 3D and 2D, (**E**,**F**) Interaction of acarbose visualized in 3D and 2D, respectively (coloured: bound residues, violet: surrounding residues).

**Figure 6 molecules-27-01888-f006:**
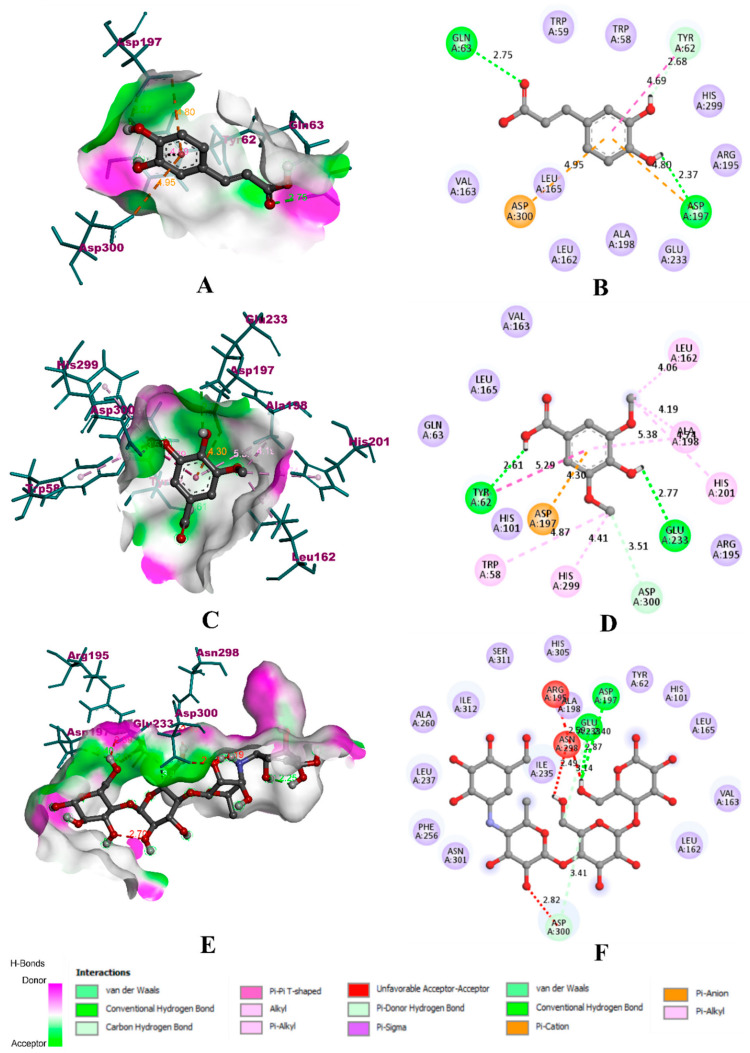
Visualization of docking simulation of experimental compounds with α-amylase. (**A**,**B**) Interaction of caffeic acid visualized in 3D and 2D, (**C**,**D**) Interaction of syringic acid visualized in 3D and 2D, (**E**,**F**) Interaction of acarbose visualized in 3D and 2D, respectively (coloured: bound residues, violet: surrounding residues).

**Figure 7 molecules-27-01888-f007:**
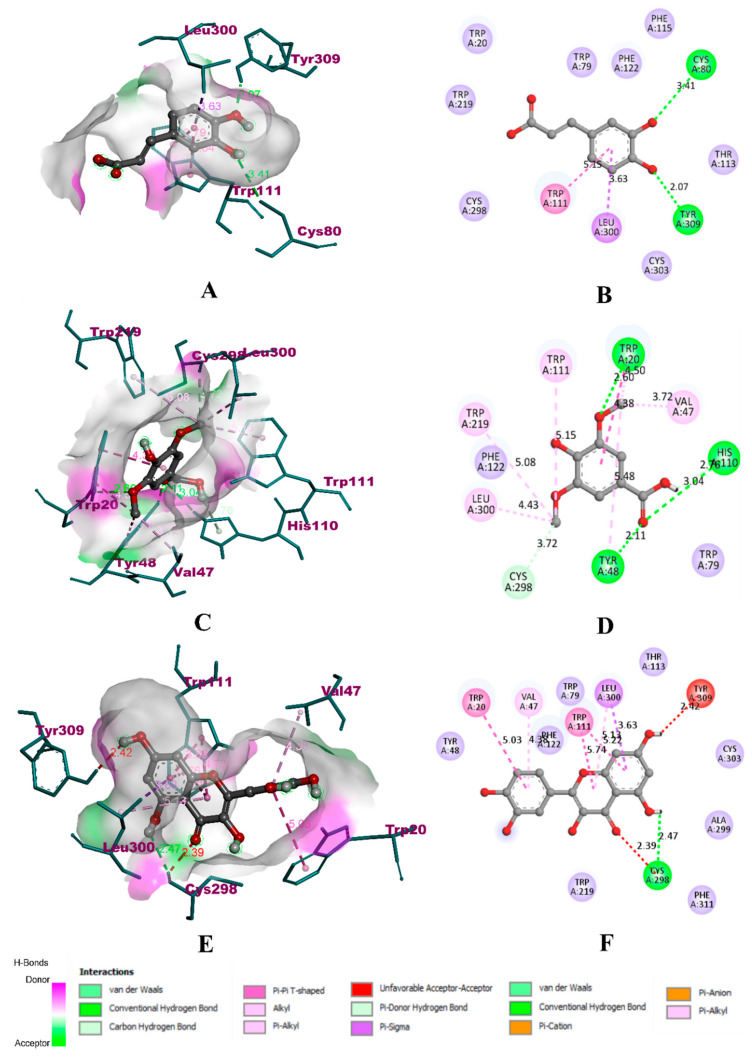
Visualization of docking simulation of experimental compounds with HAR. (**A**,**B**) Interaction of caffeic acid visualized in 3D and 2D, (**C**,**D**) Interaction of syringic acid visualized in 3D and 2D, (**E**,**F**) Interaction of quercetin visualized in 3D and 2D, respectively (coloured: bound residues, violet: surrounding residues).

**Figure 8 molecules-27-01888-f008:**
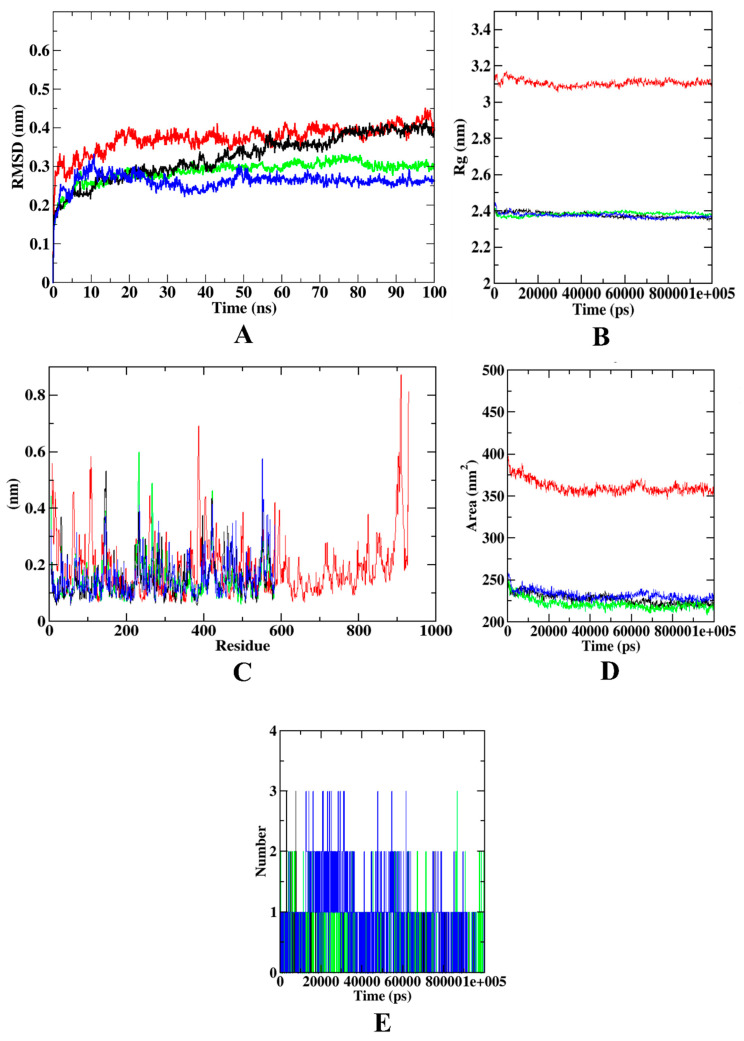
Visualization of dynamics simulation of experimental compounds with α-glucosidase. (**A**) protein–ligand complex RMSD, (**B**) Rg, (**C**) RMSF, (**D**) SASA, (**E**) ligand hydrogen bonds. Red: protein backbone atoms, green: protein–caffeic acid complex, black: protein–syringic acid complex, blue: protein–acarbose complex.

**Figure 9 molecules-27-01888-f009:**
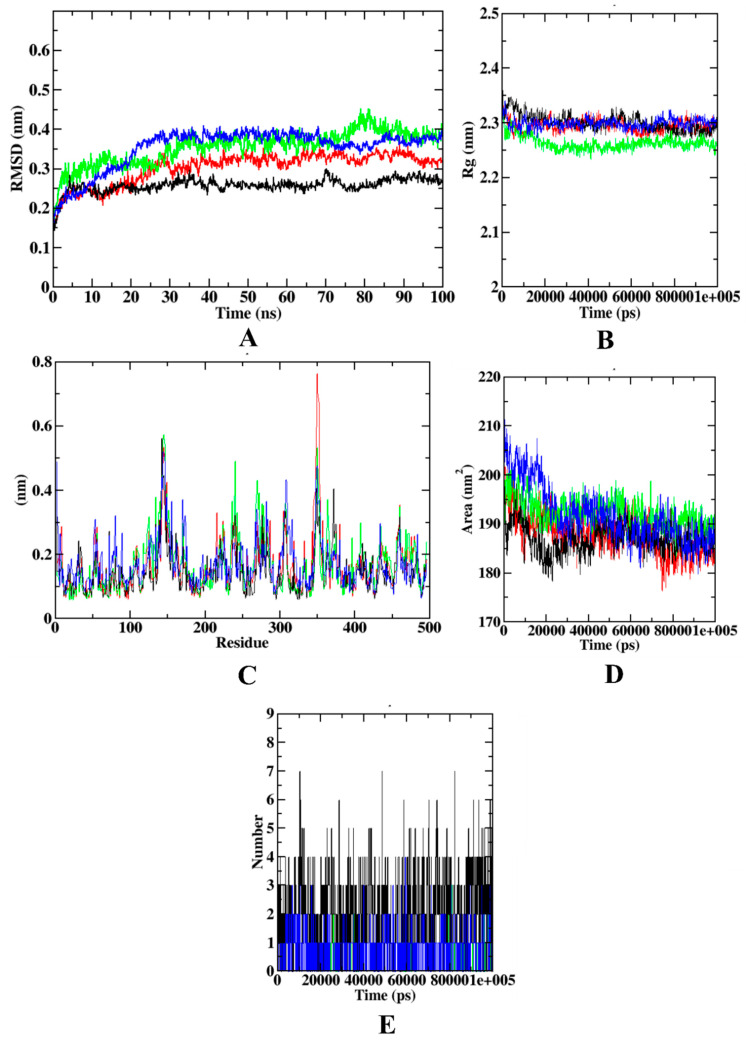
Visualization of dynamics simulation of experimental compounds with α-amylase. (**A**) protein–ligand complex RMSD, (**B**) Rg, (**C**) RMSF, (**D**) SASA, (**E**) ligand hydrogen bonds. Red: protein backbone atoms, green: protein–caffeic acid complex, black: protein–syringic acid complex, blue: protein–acarbose complex.

**Figure 10 molecules-27-01888-f010:**
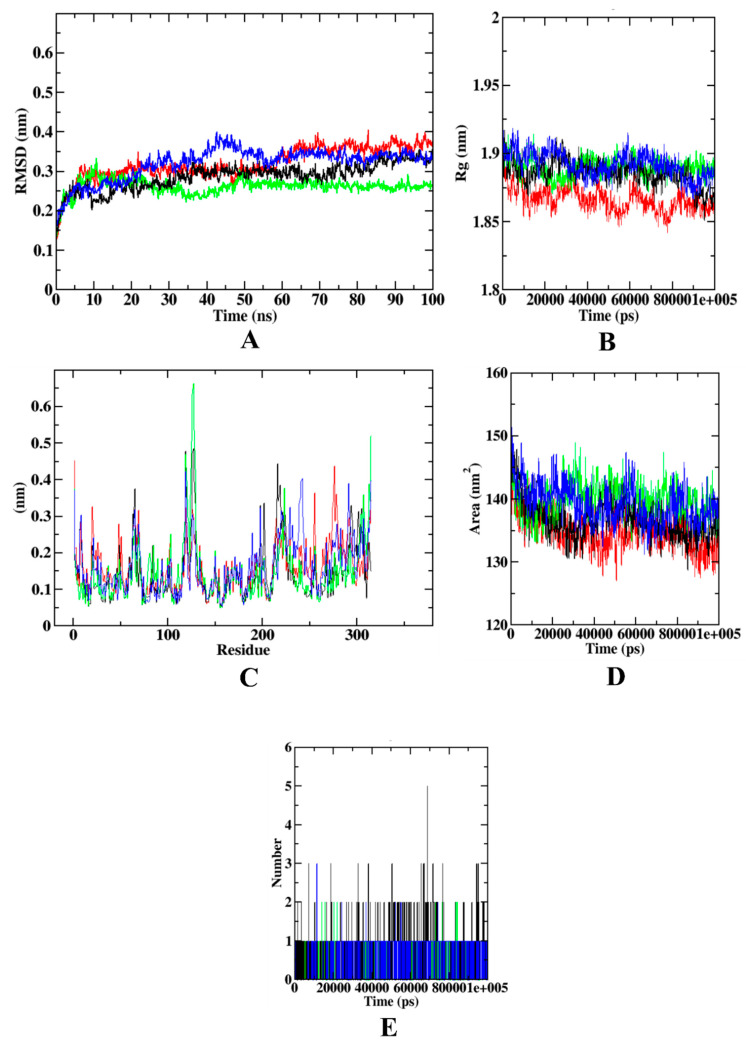
Visualization of dynamics simulation of experimental compounds with HAR. (**A**) protein–ligand complex RMSD, (**B**) Rg, (**C**) RMSF, (**D**) SASA, (**E**) ligand hydrogen bonds. Red: protein backbone atoms, green protein–caffeic acid complex, black: protein–syringic acid complex, blue: protein–quercetin complex.

**Table 1 molecules-27-01888-t001:** Phenolic compounds and ascorbic acid profiling of methanol extract of green jackfruit flour (MJ) by HPLC analysis.

Sl. No.	Name	Ret. Time	Area	Height	Concentration (µg/mg) in MJ
1	Ascorbic acid	4.222	72,576	8335	6.828
2	Gallic acid	5.708	65,639	11,867	6.176
3	Methyl gallate	12.675	60,233	5868	5.667
4	Caffeic acid	15.060	22,368	2486	2.104
5	Syringic acid	15.451	23,443	2626	2.206
6	Ferulic acid	22.084	11,293	905	1.062
7	Quercetin	29.964	6706	787	0.631
8	Kaempferol	34.914	54,923	5186	5.167

**Table 2 molecules-27-01888-t002:** Inhibitory potential of methanol extract of green jackfruit flour (MJ), ethyl acetate fraction (EaFr.) and its isolated compounds (caffeic acid and syringic acid) against α-amylase, α-glucosidase and aldose reductase enzymes.

Enzymes	IC_50_ ^x,y^ (µg/mL)				
	MJ	EaFr.	Caffeic Acid	Syringic Acid	Acarbose/* Quercetin
α-Amylase	28.00 ± 0.03 ^d^	27.80 ± 0.06 ^c^	26.90 ± 0.05 ^b^	25.25 ± 1.00 ^a^	28.50 ± 0.05 ^d^
α-Glucosidase	10.00 ± 0.14 ^b^	09.55 ± 0.87 ^b^	8.00 ± 0.40 ^a^	7.50 ± 1.05 ^a^	11.00 ± 0.11 ^c^
Aldose reductase	3.75 ± 0.75 ^b,c^	3.60 ± 0.00 ^b^	3.10 ± 0.33 ^a^	3.00 ± 0.00 ^a^	4.10 ± 0.22 ^c,^*

^x^ Values are reported as mean ± SE. Duncan multiple range test shows that means in the same row with different superscript letters ^a–d^ are significantly different (*p* ≤ 0.05). ^y^ The inhibitor concentration required to block 50% of enzyme activity is known as the IC_50_ value under assay conditions. * In the aldose reductase assay, quercetin was employed as a positive control.

**Table 3 molecules-27-01888-t003:** Enzyme kinetics of caffeic acid (CA) and syringic acid (SA) against α-amylase, α-glucosidase and aldose reductase enzymes.

Enzymes	Compound	Treatment	Mode of Inhibition ^x^	Km (µM)	Vmax (10^3^(µM/min)^−1^	K_i_ (mg) ^y,z^
α-Amylase	CA	Control	Competitive	0.79	25.25	1.03 ± 0.15
		IC_20_ 10.75 µg		0.91	26.15	
		IC_40_ 21.50 µg		1.82	25.75	
		IC_60_ 32.25 µg		3.02	26.10	
	SA	Control	Competitive	0.94	50.55	1.25 ± 0.04
		IC_20_ 10.10 µg		1.55	51.10	
		IC_40_ 20.20 µg		2.88	51.11	
		IC_60_ 30.30 µg		3.99	50.95	
α-Glucosidase	CA	Control	Competitive	1.01	40.05	0.52 ± 0.02
		IC_20_ 3.20 µg		2.44	39.98	
		IC_40_ 6.40 µg		3.33	40.40	
		IC_60_ 9.60 µg		5.01	40.37	
	SA	Control	Competitive	1.22	34.00	0.96 ± 0.22
		IC_20_ 3.0 µg		3.58	33.55	
		IC_40_ 6.0 µg		5.54	33.43	
		IC_60_ 9.0 µg		8.80	33.79	
Aldose reductase	CA	Control	Non-competitive	0.79	27.71	1.11 ± 0.36
		IC_20_ 1.25 µg		0.81	13.35	
		IC_40_ 2.50 µg		0.81	10.15	
		IC_60_ 3.75 µg		0.80	5.05	
	SA	Control	Non-competitive	0.81	78.60	1.64 ± 0.65
		IC_20_ 1.20 µg		0.91	34.47	
		IC_40_ 2.40 µg		0.92	20.05	
		IC_60_ 3.60 µg		0.92	11.50	

^x^ inhibition mode was performed from double reciprocal plot. ^y^ K_i_ = dissociation constant. ^z^ Values are expressed as mean ± SE.

**Table 4 molecules-27-01888-t004:** Total phenolic content (TPC), total flavonoid content (TFC), and antioxidant activity of methanol extract of green jackfruit flour (MJ), ethyl acetate fractions (EaFr.) and its isolated compounds (caffeic acid and syringic acid).

Sample	TPC (mg GAE/g)	TFC (mg QE/g)	EC_50_ ^x,y^ (μg/mL)		
			Radical Scavenging Activities		
			DPPH	ABTS	Superoxide
MJ	252.07 ± 0.15 ^b^	601.05 ± 0.24 ^b^	24.30 ± 0.82 ^c^	20.80 ± 1.32 ^c^	44.50 ± 2.40 ^c^
EaFr.	153.75 ± 0.36 ^a^	365.04 ± 2.00 ^a^	24.02 ± 1.87 ^c^	20.01 ± 0.33 ^c^	44.06 ± 1.78 ^c^
Caffeic acid	-		18.50 ± 0.08 ^b^	12.44 ± 1.60 ^b^	30.13 ± 2.05 ^b^
Syringic acid	-		16.00 ± 0.13 ^a^	11.40 ± 2.04 ^a^	28.00 ± 1.19 ^a^
BHA	-	-	40.25 ± 0.30 ^d^	31.00 ± 0.55 ^d^	64.75 ± 0.13 ^d^

^x^ Values are reported as mean ± SE. Duncan multiple range test shows that means in the same column with different superscript letters ^a–d^ are significantly different (*p* ≤ 0.05). ^y^ The effective concentration required to exhibit 50% of antioxidant activity is known as the EC_50_ value under assay conditions.

**Table 5 molecules-27-01888-t005:** After 3 weeks of incubation, the effects of methanol extract of green jackfruit flour (MJ) and its separated components on Fructosamine, Protein Carbonyl, and Thiols group concentration in the HSA/Fructose system.

**A.**	**Fructosamine (mmol/mg Protein)**						
	Week	HSA	HSA/Fructose	MJ	Caffeic Acid	Syringic Acid	Aminoguanidine
	1	-	42.15 ± 1.06 ^e^	27.15 ± 0.31 ^c^	26.82 ± 0.14 ^a^	26.30 ± 1.21 ^b^	28.50 ± 1.16 ^d^
	2	-	54.08 ± 0.18 ^e^	28.00 ± 1.10 ^c^	27.06 ± 0.13 ^a^	26.99 ± 1.90 ^b^	30.00 ± 1.00 ^d^
	3	-	60.50 ± 1.06 ^e^	28.80 ± 1.72 ^c^	28.02 ± 1.04 ^a^	26.41 ± 1.18 ^b^	31.75 ± 0.97 ^d^
**B.**	**Protein carbonyl Content (nmol/mg protein)**						
	1	0.50 ± 0.20 ^a^	2.06 ± 2.12 ^f^	0.57 ± 0.12 ^d^	0.49 ± 2.17 ^b^	0.49 ± 1.06 ^c^	0.64 ± 0.54 ^e^
	2	0.52 ± 0.14 ^a^	4.20 ± 0.00 ^f^	0.58 ± 0.02 ^d^	0.57 ± 1.25 ^b^	0.54 ± 0.62 ^c^	0.66 ± 1.05 ^e^
	3	0.55 ± 0.27 ^a^	6.66 ± 0.76 ^f^	0.58 ±1.44 ^d^	0.57 ± 2.22 ^b^	0.54 ± 1.32 ^c^	0.70 ± 1.11 ^e^
**C.**	**Thiols Group (nmol/mg protein)**						
	1	2.20 ± 1.16 ^d^	1.64 ± 0.14 ^c^	0.76 ± 0.57 ^b^	0.69 ± 0.55 ^a^	0.64 ± 0.62 ^a^	0.95 ± 1.46 ^b^
	2	2.45 ± 1.32 ^d^	1.95 ± 0.31 ^c^	0.79 ± 0.48 ^b^	0.79 ± 0.05 ^a^	0.76 ± 0.53 ^a^	1.04 ± 1.00 ^b^
	3	2.99 ± 0.42 ^d^	2.04 ± 1.04 ^c^	0.81 ± 0.98 ^b^	0.80 ± 0.88 ^a^	0.78 ± 0.34 ^a^	1.17 ± 0.89 ^b^

Values are reported as mean ± SE. Duncan multiple range test shows that means in the same row with different superscript letters ^a–f^ are significantly different (*p* ≤ 0.05).

**Table 6 molecules-27-01888-t006:** Summary of virtual screening of the experimental compounds.

Compound	α-Glucosidase			α-Amylase			HAR		
	BA	NB	HB	BA	NB	HB	BA	NB	HB
Ascorbic acid	−7.6	4	0	−6.8	5	1	−4.2	7	0
Gallic acid	−5.3	7	2	−5.2	7	2	−6.3	6	2
Methyl gallate	−4.5	2	1	−6.1	5	0	−7.1	9	2
Caffeic acid	−8.2	6	4	−8.1	6	3	−7.4	4	2
Syringic acid	−11.4	10	4	−12.5	11	3	−12.9	10	4
Ferulic acid	−5.2	5	1	−6.7	7	1	−8.9	8	3
Quercetin	−6.2	7	2	−7.1	6	3	−10.3	7	1
Kaempferol	−4.7	2	0	−8.1	6	2	−9.1	9	3
Acarbose	−10.2	7	6	−6.2	2	2	-	-	-

BA: binding affinity in kcal/mol, NB: Total number of non-bonding interactions, HB: Total number of hydrogen bonds, HAR: human aldose reductase.

**Table 7 molecules-27-01888-t007:** Druglikeness and pharmacokinetics of experimental compounds.

Categories	Parameters	Caffeic Acid	Syringic Acid	Acarbose	Amino-Guanidine	Quercetin
Druglikeliness	Mol. Wt.	180.04	198.05	645.25	74.06	302.04
	nHA	4	5	19	4	7
	nHD	3	2	14	6	5
	TPSA	77.76	75.99	321.17	87.92	131.36
	LogP	1.43	1.212	−4.37	−2.376	2.155
Absorption	Caco-2	−5.22	−5.142	−6.149	−5.448	−5.204
	MDCK	1.1	1.1	0.00089	0.001687	8.0
Distribution	VD	0.37	0.259	0.071	0.918	0.579
	BBB	0.119	0.457	0.385	0.361	0.008
Metabolism	CYP1A2	0.048	0.032	0.0	0.029	0.943
	CYPC19	0.069	0.025	0.002	0.025	0.053
	CYP2C9	0.036	0.028	0.0	0.011	0.598
	CYP2D6	0.014	0.012	0.0	0.011	0.411
	CYP3A4	0.043	0.016	0.0	0.006	0.348
Excretion	Clearance	10.973	7.208	0.373	5.857	8.284
Toxicity	hERG	0.018	0.034	0.04	0.0051	0.099
	AMES	0.183	0.009	0.0099	0.875	0.657

Mol. Wt.: Molecular weight in g/mol (optimal 100–600), nHA: number of hydrogen bond acceptors (optimal 0–12), nHD: number of hydrogen bond donors (optimal 0–7), TPSA: topological polar surface area (optimal 0–140), LogP: Log of the octanol/water partition coefficient (optimal 0–3), Caco-2: Caco-2 cell permeability (optimal > −5.15), MDCK: Madin−Darby Canine Kidney cells permeability (optimal: >2 × 10^−6^ cm/s), VD: volume distribution in L/kg (optimal 0.04–20), BBB: blood–brain barrier (optimal 0–0.3), CYP: cytochrome P (optimal near to 0.0), clearance in mL/min/kg (optimal > 5), hERG: human ether-à-go-go gene (optimal 0–0.3), AMES mutagenicity (optimal 0–0.3).

**Table 8 molecules-27-01888-t008:** Binding free calculations of experimental molecules complexed with target proteins.

Protein–Ligand Complexes	Types of Binding Free Energies					
	Values and Standard Deviations	Van der Waal’s Energy	Electrostatic Energy	Polar Solvation Energy	SASA Energy	Binding Energy
α-Glucosidase- caffeic acid	Values (KJ/mol)	−76.593	−28.312	56.039	−7.771	−56.637
	Standard deviation (KJ/mol)	+/−86.888	+/−37.350	+/−63.731	+/−8.772	+/−72.011
α-Glucosidase- syringic acid	Values (KJ/mol)	−170.549	−17.803	50.133	−13.855	−152.074
	Standard deviation (KJ/mol)	+/−85.257	+/−10.149	+/−31.329	+/−7.492	+/−78.335
α-Glucosidase- acarbose	Values (KJ/mol)	−131.001	−6.710	58.293	−10.168	−89.586
	Standard deviation (KJ/mol)	+/−177.536	+/−9.451	+/−61.431	+/−13.817	+/−158.089
α-Amylase- caffeic acid	Values (KJ/mol)	−29.394	−0.791	4.329	−2.644	−26.918
	Standard deviation (KJ/mol)	+/−66.039	+/−4.572	+/−34.555	+/−6.106	+/−57.630
α-Amylase- syringic acid	Values (KJ/mol)	−109.781	−33.898	52.824	−9.420	−100.275
	Standard deviation (KJ/mol)	+/−102.373	+/−34.252	+/−86.454	+/−8.557	+/−83.569
α-Amylase- acarbose	Values (KJ/mol)	−122.109	−30.198	42.314	−10.521	−90.275
	Standard deviation (KJ/mol)	+/−102.373	+/−24.152	+/−56.245	+/−7.522	+/−63.569
Human aldose reductase-caffeic acid	Values (KJ/mol)	−84.938	−21.601	37.285	−7.192	−76.445
	Standard deviation (KJ/mol)	+/−62.813	+/−25.688	+/−32.088	+/−5.183	+/−62.792
Human aldose reductase-syringic acid	Values (KJ/mol)	−149.669	−6.992	79.945	−12.899	−109.615
	Standard deviation (KJ/mol)	+/−101.479	+/−11.374	+/−50.793	+/−7.329	+/−73.901
Human aldose reductase-quercetin	Values (KJ/mol)	−159.669	−30.870	82.920	−13.796	−100.299
	Standard deviation (KJ/mol)	+/−100.389	+/−26.576	+/−72.468	+/−11.937	+/−98.464

## Data Availability

Not applicable.

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
