# Peer review of "Inhibitory Effect of Polyphenols from the Whole Green Jackfruit Flour against α-Glucosidase, α-Amylase, Aldose Reductase and Glycation at Multiple Stages and Their Interaction: Inhibition Kinetics and Molecular Simulations"

_molecules, 2022, doi:10.3390/molecules27061888_

Round 1
Reviewer 1 Report
Tejaswini and colleagues conducted research to evaluate the composition and biological potential of a plant product. They used classical laboratory techniques for isolating components, but also more modern techniques for spectral, enzymatic or silico characterization.
The paper is largely acceptable, but a number of issues need to be corrected.
In many places in paper "ml" must be replaced with "mL".
The in text citation must be uniformized in paper (... by Seal (2016) and in another place is ... Ramu et al. 2014. )
The Discussion section is a little too big.
The biggest concern is regarding the interaction of the compounds with HSA (docking and molecular dynamics). The authors must explain how the interaction of the phenolic compounds with HSA would influence the chemical stability (non-reactivity) of the protein in a sugar environment. The Amadori reaction is a pure chemical reaction between amine and carbonyl compound. In my opinion is has nothing to do with the binding of ligands to the protein. Personal, I find the compounds-HSA interaction study useless.
"we established the antiglycaemic potential" is wrong. The present research has nothing to do with glicaemia.
Please give details about the system used for the molecular dynamics simulation (GPU/CPU, operating system or virtual)
3.10 about acarbose and Lipinski. Authors are wrong. Acarbose is non-absorbed from GI. It must stay in GI to perform the inhibition of the enzyme. So, all discussions about its pharmacokinetics must be removed, because it is useless.
Please give supplementary data about quercetin carcinogenicity.
Reviewer 2 Report
Regarding manuscript 1597673 entitled “Inhibitory effect of polyphenols from the whole green jackfruit flour against α – glucosidase, α – amylase, aldose reductase & glycation at multiple stages and their interaction: Inhibition kinetics and molecular stimulations", the manuscript is interesting, however there are some critical points which need to be clarify and improved to make the manuscript suitable for publication.
- First, it is not clear how the choice of fractions was made during purification. It seems that is only about the antioxidant activity, or not? In the abstract part the authors refer to a “bioassay-guided separation of the active antihyperglycaemic compounds”. Thus, in order to identify the extract or fractions with the best inhibitory activity, the bioassay-guided separation should be done following the enzyme inhibitory activity of the fractions.
- Moreover, why caffeic acid and are isolated? The fractions in which they are present were the most active against the enzymatic activities? In this case, the data of inhibition of every single fraction should be reported.
- Figure 2: Since the authors report the Lineweaver-Burk plot constructed by plotting 1/enzyme activity (1/v) versus 1/substrate (1/[S]) (Line 191 and Figure 2), why the authors used the Dixon plot to determine the ki values? They can derive the ki values from secondary plots.
- Abstract: I suggest to remove the sentence “The peel of jackfruit (Artocarpus heterophyllus L.) is one of the least exploited by-product 18 in both jackfruit cultivation and processing” because it seems that the paper is focused on the peel of jackfruit, indeed it is on whole jackfruit.
- Lines 101-105: “The methanol extract of whole jackfruit flour (MJ) was subjected to preliminary phytochemical screening to determine the phytoconstituents present, according to standard protocols [9]. The total flavonoid content (TFC) and total phenol content (TPC) for MJ were determined as per Ordon et al. (2006) [10] and Shuxia et al. (2013) [11], respectively”. I think that It is better to remove this part from this paragraph which is about “extraction”.
- Line 135. There is a period missing at the end of the sentence.
- Line 190: why the different concentration of compounds are indicated as “IC20, IC40 and IC60”?
- Line 294: it refers to Table 2 and not Table 1.
- Line 297: there is an “alpha” missing before –amylase.
- Figure 2: Please check the symbol and the concentration because there are mistakes in panel A and D.
- Table 1: change “Ferrulic acid” with “Ferulic acid”.
- Table 2: Please put Table 2 before Figure 2 in the text. In fact, Table 2 is cited before in the text on respect of Figure 2.
- Table 3: The values of IC50 are micromolar and not mM, right? Please change it in the table.
- The “z” seems to be not present in the table. Please check it.
- Table 4: I don’t understand the meaning of measure the total phenolic content (TPC) of caffeic acid and Syringic acid since they are isolated compounds.
Reviewer 3 Report
The manuscript “Inhibitory effect of polyphenols from the whole green jackfruit flour against α – glucosidase, α – amylase, aldose reductase & glycation at multiple stages and their interaction: Inhibition kinetics and molecular simulations” describes isolation, identification and inhibitory effects of few polyphenols whole jackfruit flour (MJ) focussing on caffeic acid and syringic acid. Experimental studies are complemented with in silico studies based on molecular modeling, docking and dynamics calculations.
The topic is interesting however computational methodology needs improvement before considering the manuscript for publication.
- Molecular modelling results are missing. Model validation, based on various tools has to be reported
- Docking and scoring can be used to predict binding modes and discriminate between binders and non-binders, but are not particularly accurate in determining ligand-binding affinities. I suggest to use MM/PBSA and MM/GBSA methods to estimate the free energy of binding. Lines 796-797: “Results from the docking simulation reveal that caffeic acid and syringic acid showed the highest degree of inhibition”. This sentence has to be re-written or removed. What does it mean?
- MD simulations are rarely performed singly in recent years. I suggest to run simulations in triplicate (reporting supporting MD simulations in the Supplementary)
- 807-810 and further on: Analysis of interactions to assess the catalytic mechanism has to be supported plotting the described interactions as function of time in the MD results
- English of the computational results needs extensive revision
Minor points:
- 197: Please specify the Uniprot code
- 340: define C1 and C2
- Figure 8: please explain the RMSF trend from residue 600 to about 1000
- Title and keywords: correct stimulations into simulations
Round 2
Reviewer 1 Report
Authors made significant improvements to the paper.
Some corrections need to be performed:
-table 1. Phenolic compounds....
but in table is present ascorbic acid. Please change the description of the table
-line 228: Intel 1165G7 CPU is intended for mobile devices. The authors presented the device used as "desktop workstation". Please clarify
Author Response
Response to reviewer comments:
All the authors thank the reviewer for the thoughtful suggestions to improve the quality of
the manuscript. The response to all the comments point-by-point are provided below.
Reviewer 1 comments
-table 1. Phenolic compounds.... but in table is present ascorbic acid. Please change the description of the table
Authors’ response: As per the reviewer’s suggestion, the Table 1 caption has been changed to “Phenolic compounds and ascorbic acid profiling of methanol extract…..” to be more precise.
The authors had performed the HPLC analysis and investigated the ascorbic acid profile but had not included in the Table 1 caption and now it has been incorporated for more clarity.
-line 228: Intel 1165G7 CPU is intended for mobile devices. The authors presented the device used as "desktop workstation". Please clarify.
Authors’ response: As per the reviewer’s suggestion, details about the system used for MD simulations have been incorporated in the manuscript under methodology section. We apologise for the typological error, where the details of the spare system was given initially.
Reviewer 2 Report
The authors have modified the manuscript following the previous suggestions. Thus, the manuscript can be accepted.
Only a minor suggestion:
In table 4: as previously suggested for total phenolic content, also the flavonoid content (TFC) of caffeic acid and Syringic acid should be deleted as they are single compounds.
Author Response
Response to reviewer comments:
All the authors thank the reviewer for the thoughtful suggestions to improve the quality of
the manuscript. The response to all the comments point-by-point are provided below.
Reviewer 2 comments
In table 4: as previously suggested for total phenolic content, also the flavonoid content (TFC) of caffeic acid and Syringic acid should be deleted as they are single compounds.
Authors’ response: As per the reviewer’s suggestion, the TFC (total flavonoid content) of single compounds caffeic acid and Syringic acid has been removed/deleted in the Table 4.
Reviewer 3 Report
Authors properly answered raised question but Fig S7 has to be modified showing (distance in Angstrom) the evolution of specific hydrogen bonds/distances (those indicated in the manuscript involved in the catalytic mechanism) as a function of time. This would support the catalytic mechanism.
Author Response
Response to reviewer comments:
All the authors thank the reviewer for the thoughtful suggestions to improve the quality of
the manuscript. The response to all the comments point-by-point are provided below.
Reviewer 3 comments
Authors properly answered raised question but Fig S7 has to be modified showing (distance in Angstrom) the evolution of specific hydrogen bonds/distances (those indicated in the manuscript involved in the catalytic mechanism) as a function of time. This would support the catalytic mechanism.
Authors’ response: As per the reviewer’s suggestion, Fig S7 has been modified showing (distance in Angstrom) the evolution of specific hydrogen bonds/distances (those indicated in the manuscript involved in the catalytic mechanism) as a function of time.